# EFFL: Egalitarian Fairness in Federated Learning for Mitigating Matthew Effect

## Abstract

Recent advances in federated learning (FL) enable collaborative training of machine learning (ML) models from large-scale and widely dispersed clients while protecting their privacy. However, when different clients' datasets are heterogeneous, traditional FL mechanisms produce a global model that does not adequately represent the poorer clients with limited data resources, resulting in lower accuracy and higher bias on their local data. According to the **Matthew effect**, which describes how the advantaged gain more advantage and the disadvantaged lose more over time, deploying such a global model in client applications may worsen the resource disparity among the clients and harm the principles of social welfare and fairness. To mitigate the Matthew effect, we propose **Egalitarian Fairness Federated Learning** (EFFL), where **egalitarian fairness** refers to the global model learned from FL has: ❶ equal accuracy among clients; ❷ equal decision bias among clients. Besides achieving egalitarian fairness among the clients, EFFL also aims for performance optimality, minimizing the empirical risk loss and the bias for each client; both are essential for any ML model training, whether centralized or decentralized. We formulate EFFL as a multi-constrained multi-objectives optimization (MCMOO) problem, with the decision bias and egalitarian fairness as constraints and the minimization of the empirical risk losses on all clients as multiple objectives to be optimized. We propose a gradient-based three-stage algorithm to obtain the Pareto optimal solutions within the constraint space. Extensive experiments demonstrate that EFFL outperforms other state-of-the-art FL algorithms in achieving a high-performance global model with enhanced egalitarian fairness among all clients.

## 1 Introduction

Federated learning (FL) (McMahan et al., 2017) has emerged as a significant learning paradigm in which clients utilize their local data to train a global model collaboratively without sharing data. FL has attracted wide attention from various fields, especially in domains where data privacy and security are critical, such as healthcare, finance, and social networks. However, when the data distribution among clients is heterogeneous, the global model may perform inconsistently across different clients. This raises a **client-level** fairness issue: how to define and achieve a fair model performance for each client. From the perspective of commercial or profit-driven clients, **contribution fairness** (Tay et al., 2022; Liu et al., 2022) is attractive, which requires that the model performance on each client is proportional to their data resource contribution to the FL model.

In the real-world, clients may have unequal data resources due to historical or unavoidable factors. They deserve fair treatment based on social welfare and equality principles. However, *contribution fairness* worsens the resource plight of poorer clients. For instance, when hospitals with non-i.i.d. datasets collaborate to train a disease diagnosis model, the hospitals with lower data resources will receive a model that does not fit well with their data distribution, as high-resource hospitals more dominate the model optimization. Therefore, the local model performance in a low-resource hospital may exhibit uncertain accuracy and decision fairness. Such a low-trustworthy model may affect the subsequent diagnosis of low-resource clients, leading to persistent resource inequality and the deterioration of social welfare. This phenomenon is referred to as the **Matthew effect** (Merton, 1968), a social psychological phenomenon that describes how the rich get richer and the poor get poorer in terms of resources such as education, economy, and data information, etc.

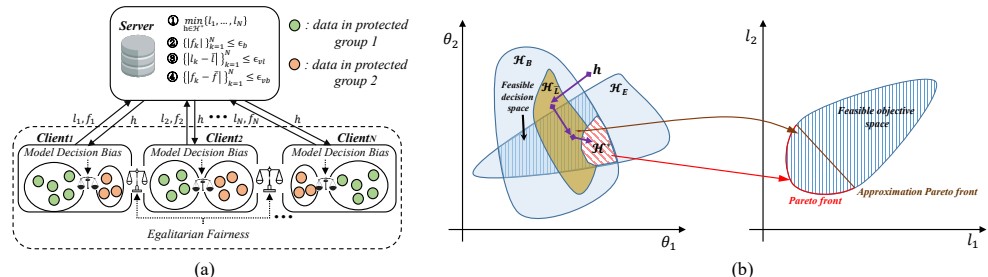

Figure 1: (a) Training goals in EFFL; (b) Optimization paths to achieve a Pareto solution $h \in \mathcal{H}^*$ for EFFL.

To mitigate the Matthew effect, we define **egalitarian fairness** in FL settings, which requires that the global model exhibits equal performance across clients. We consider two aspects of equal performance: ❶ Accuracy: accuracy reflects how well the global model fits the local data of clients; thus equal accuracy performance can enhance the poor-clients decision quality; ❷ Decision bias: decision bias reflects how fair of the global model decision on different protected groups such as gender, race, or religion within the client, thus equal decision bias performance can enhance the poor-clients reputation and decision credibility.

**Attaining egalitarian fairness in FL settings presents significant challenges.** ❶ First, egalitarian fairness requires equal performance across all clients, which creates a trade-off with the general objective of obtaining highest performance during model training. This trade-off is especially evident when dealing with heterogeneous clients, where the global model exhibits varied performance. For example, an advantaged client must sacrifice local performance for egalitarian fairness, because exceptional high local performance may result from the model's poor performance on other clients, which is undesirable from the standpoints of social welfare and ethics. Our goal is to strike a balance in this trade-off so that all clients achieve high and equitable performance; ❷ The second challenge is that heterogeneous local datasets can lead to conflicting gradient descent directions, posing the trade-off that improving the performance for one client could potentially degrade the performance of others; ❸ Furthermore, as we consider egalitarian fairness in both accuracy and decision bias, it is noted that there is a potential trade-off between accuracy and decision bias (Wang et al., 2021).

Several works have focused on client-level fair model training in FL. Mohri et al. (2019) proposed a min-max-based approach that prioritized the model training optimization towards the worst-performing client. However, they focused on improving the performance of the worst-performing client and may harm others, thus failed to achieve performance equality, especially when there were more than two clients. Cui et al. (2021) simultaneously minimized the empirical risk loss across all clients. Li et al. (2021) allowed clients to fine-tune the global model on local data. These methods enhance the model's representation of all clients but cannot guarantee to narrow the gap between the better-performing and the worse-performing clients. To achieve equal model performance, improving the model fitting towards the data distribution of the poorer clients is necessary. Li et al. (2019) proposed q-FFL, which enhances the weights of the poorer clients in the aggregation process to achieve equal accuracy across the clients. However, the method is heuristic and depends on a hyperparameter $q$, which is difficult to be tuned to achieve the optimal result.

**Our main contributions include:** ❶ We propose an FL framework, called **Egalitarian Fair Federated Learning** (EFFL), that aims to achieve an egalitarian fair global model that provides both high performance and egalitarian fairness for clients; ❷ We formally define the learning goals in EFFL and form it as a multi-constrained multi-objectives optimization (MCMOO) problem. As shown in Fig. 1 (a), the objectives are to minimize the local empirical risk losses $\{l_1, ..., l_N\}$ on $N$ clients (goal 1). The local decision biases $\{f_1, ..., f_N\}$ on $N$ clients are constrained to be below an acceptable bias budget (goal 2). These two goals jointly ensure higher model performance across clients. We impose constraints on the deviations of local empirical risk loss and local decision bias from their mean values (goal 3 and goal 4) to achieve egalitarian fairness with equal performance among clients; ❸ To address the challenges mentioned above, we propose a three-stage gradient-based algorithm that achieves Pareto optimal solutions in the decision space defined by the decision bias and egalitarian fairness constraints, where the global model performance on each local client is maximized and cannot be further improved without harming others; ❹ We perform comprehensive

experiments on both synthetic and real-world datasets and show that our proposed EFFL framework can achieve a global model that outperforms state-of-the-art (SOTA) baselines in terms of both performance and egalitarian fairness across clients.

## 2 RELATED WORK

Most of the existing fairness research (Roh et al., 2021; Shen et al., 2022; Choi et al., 2021; Sankar et al., 2021) assumes that the training process can access the whole training dataset in a centralized manner. However, this assumption does not hold in real-world scenarios where data privacy and communication constraints prevent clients from sharing their data with a central server. In FL, several work (Ezzeldin et al., 2023; Papadaki et al., 2022; Chu et al., 2021; Du et al., 2021; Cui et al., 2021; Hu et al., 2022) aims to reduce model outcome bias towards different protected groups such as gender, race, or age. Recently, there has been a growing interest in achieving fairness towards clients in FL. Client-level fairness is equivalent to the existing group-level fairness when a client exclusively contains one, and only one, protected group. However, this is a rare case. Therefore, it is necessary to design client-level fair FL to cover more general scenarios. Mohri et al. (2019) propose AFL, a min-max optimization scheme that focuses on the worst-performing client. Li et al. (2021) propose Ditto to achieve training fairness by allowing clients to fine-tune the received global model using local data. Cui et al. (2021) propose FCFL to jointly consider accuracy consistency and decision bias across different local clients (data sources) by minimizing the model loss of the worst-performing client. These works enhance a fair representation of different clients during training but cannot guarantee equal performance. To achieve equal accuracy across clients, Li et al. (2019) propose q-FFL, a heuristic method that adjusts the weights of the aggregate process to enhance the influence of poorer individuals. Pan et al. (2023) propose FedMDFG that adds cosine similarity between the loss vectors among clients and the unit vector as a fairness objective in the local loss functions. Previous work overlooks the trade-offs in achieving equality from a social welfare perspective and local optimality from an individual beneficial perspective. Moreover, as there is an inherent trade-off between accuracy and decision bias for each client (Wang et al., 2021), we also need to consider the deterioration of the decision bias caused by accuracy distribution adjustment. It is necessary to ensure that the decision bias of each individual is within an acceptable budget to provide a trustworthy global model. For the same social welfare purpose as accuracy equality, maintaining decision bias equality among individuals is helpful to improve the reputation and decision credibility of poor clients. We argue that optimizing the accuracy objective alone is insufficient for an FL system that adheres to social ethics and legal regulations. Therefore, we introduce a novel FL framework, EFFL, to produce a global model with high and equitable performance across clients.

## 3 PRELIMINARIES

In this section, we provide the notions and formally define the problem of **Egalitarian Fairness Federated Learning** (EFFL), which extends the fairness criteria in FL and covers novel application scenarios.

### 3.1 FEDERATED LEARNING

We focus on horizontal FL(Yang et al., 2019), which involves $N$ clients, each associated with a specific dataset $\mathcal{D}_k = \{X_k, A_k, Y_k\}$, where $k \in \{1, ..., N\}$, $X_k$ denotes the general attributes of the data without protected information, $A_k$ denote a protected attribute, such as gender, race, or religion, and $Y_k$ denoted truth label. The FL procedure involves multiple rounds of communication between the server and the clients. In each round, the server sends the global model $h_\theta$ with parameter $\theta$ to the clients, who then train their local models on their local private datasets $\{\mathcal{D}_1, ..., \mathcal{D}_N\}$, resulting in local models $\{h_{\theta_1}, ..., h_{\theta_N}\}$. The server then aggregates the local parameters and updates the global model for the next communication round (McMahan et al., 2017). The original FL (McMahan et al., 2017) aims to minimize the average empirical risk loss over all the clients' datasets, and the optimal hypothesis parameter $\theta^*$ satisfies:

$$\theta^* = \arg\ \min_{\theta \in \Theta} \sum_{k=1}^{N} \left( \frac{|\mathcal{D}_k|}{\sum_{k=1}^{N} |\mathcal{D}_k|} l_k\left(\hat{Y}_k, Y_k\right) \right),\tag{1}$$

where $\hat{Y}_k = h_\theta\left(X_k, A_k\right)$ is the output of the hypothesis $h_\theta$ when input $(X_k, A_k)$ and $l_k(\cdot)$ is the loss function for $k$-th client.

### 3.2 MODEL PERFORMANCE METRICS

As mentioned in Sec. 1, we study egalitarian fairness in terms of two aspects of model performance: accuracy and decision bias. The local accuracy of the global model on the $k$-th client is naturally controlled by the training loss $l_k$, which measures the difference between the model's decision and the truth label on the local data of the $k$-th client. In the context of classification, we use the BCELoss on the $k$-th client, $l_k(h) = -\frac{1}{|Y|} \sum_{i=1}^{|Y|} \left( Y_k^i \log\left(\hat{Y}_k^i\right) + \left(1 - Y_k^i\right) \log\left(1 - \hat{Y}_k^i\right) \right)$. Decision bias refers to the disparities in model decisions made across different groups formed by protected attributes, such as gender, race, and region. We use two decision bias metrics, namely *Accuracy Parity Standard Deviation* (APSD) and *True positive rate Parity Standard Deviation* (TPSD)(Poulain et al., 2023). Taking a binary classification problem as example, the decision bias measured by APSD or TPSD for the $k$-th client is defined as APSD: $f_k(h) = \sqrt{\frac{\sum_{i=1}^{M}\left(\Pr\left(\hat{Y}_k=1|A_k=i\right)-\mu\right)^2}{M}}$ and TPSD: $f_k(h) = \sqrt{\frac{\sum_{i=1}^{M}\left(\Pr\left(\hat{Y}_k=1|A_k=i,Y_k=1\right)-\mu\right)^2}{M}}$, where $\mu$ is the average *True Positive Rate* (TPR), or accuracy under all groups divided by the values of the protected attribute, and $M$ is the number of possible values for the sensitive attribute $A_k$. A hypothesis $h_\theta$ satisfies $\epsilon_b$-decision bias on $k$-th client if $f_k(h) \le \epsilon_b$, where $\epsilon_b$ is the predefined budget for the decision bias. The proposed egalitarian fair FL can also be applied to non-binary target variables, by replacing the BCELoss with a multi-class cross-entropy loss function, i.e., $loss = -\sum_{i=0}^{C-1} y_i \log(p_i)$, and replacing the decision bias metric with a maximum-version, APSD: $f_k(h) = max_{y\in[|Y|]}\sqrt{\frac{\sum_{i=1}^{M}\left(\Pr\left(\hat{Y}_k=y|A_k=i\right)-\mu\right)^2}{M}}$ and TPSD: $f_k(h) = max_{y\in[|Y|]}\sqrt{\frac{\sum_{i=1}^{M}\left(\Pr\left(\hat{Y}_k=y|A_k=i,Y_k=y\right)-\mu\right)^2}{M}}$, respectively.

### 3.3 EGALITARIAN FAIRNESS

Egalitarian fairness in FL refers to the model providing equal performance across clients, roughly speaking, ensuring clients have levels of performance that are all roughly comparable. Therefore, we evaluate egalitarian fairness in FL based on the degree of equality in performance. In existing work, Pan et al. (2023) measured the performance equality by the cosine similarity between the model losses on all clients $[l_1, ..., l_N]$ and the unit vector $p = \mathbf{1}$. This metric fails to distinguish each client's performance and to impose precise constraints on them, especially when the demand for performance equality is dynamic. For instance, clients may allow the violation of performance equality to be within an acceptable threshold. To avoid this, we measure the model performance equality across clients by the absolute deviation of each client's performance from the mean performance of all clients. A hypothesis $h$ satisfies $\epsilon_{vl}$-egalitarian fairness on accuracy performance and $\epsilon_{vb}$-egalitarian fairness on decision bias performance if:

$$\left|l_k(h) - \bar{l}(h)\right| \le \epsilon_{vl}, \left|f_k(h) - \bar{f}(h)\right| \le \epsilon_{vb}, k \in \{1, ..., N\}.\tag{2}$$

where $\bar{l}(h) = \frac{1}{N}\sum_{k=1}^{N} l_k(h)$ and $\bar{f}(h) = \frac{1}{N}\sum_{k=1}^{N} f_k(h)$ are the average empirical risk loss and average decision bias, respectively, and $\epsilon_{vl}$ and $\epsilon_{vb}$ are the predefined budgets for the egalitarian fairness on accuracy and decision bias, respectively.

### 3.4 EGALITARIAN FAIRNESS FEDERATED LEARNING

To achieve a global model that provides both high and equal performance across clients, we propose a novel framework called **Egalitarian Fair Federated Learning** (EFFL), in which the training goals can be formulated as a multi-constrained multi-objectives optimization (MCMOO) problem.

**Definition 1** *(Egalitarian Fairness Federated Learning) We formalize the **Egalitarian Fair Federated Learning** (EFFL) problem as follows:*

$$\min_{h \in \mathcal{H}^*} \{l_1(h), ..., l_N(h)\},$$

$$s.t. \{f_k(h)\}_{k=1}^N \leq \epsilon_b, \{|l_k(h) - \bar{l}(h)|\}_{k=1}^N \leq \epsilon_{vl}, \{|f_k(h) - \bar{f}(h)|\}_{k=1}^N \leq \epsilon_{vb}, \quad (3)$$

*where $h$ is a hypothesis from a hypothesis set $\mathcal{H}^*$.*

The MCMOO problem seeks to minimize the empirical risk losses for all clients while ensuring each client has a $\epsilon_b$-decision bias. It also satisfies $\epsilon_{vl}$-egalitarian fairness for accuracy and $\epsilon_{vb}$-egalitarian fairness for decision bias. Finding the optimal solution to the MCMOO problem is nontrivial, as the objectives may conflict. Therefore, we aim to identify the Pareto-optimal hypothesis $h$, which is not dominated by any other $h' \in \mathcal{H}$. The definitions of Pareto optimal and Pareto front are as follows:

**Definition 2** *(Pareto Optimal and Pareto Front (Lin et al., 2019)) In a multi-objective optimization problem with loss function $l(h) = \{l_1(h), ..., l_N(h)\}$, we say that for $h_1, h_2 \in \mathcal{H}$, $h_1$ is dominated by $h_2$ if $\forall i \in [N], l_i(h_2) \leq l_i(h_1)$ and $\exists i \in [N], l_i(h_2) < l_i(h_1)$. A solution $h$ is Pareto optimal if it is not dominated by any other $h' \in \mathcal{H}$. The collection of Pareto optimal solutions is called the Pareto set. The image of the Pareto set in the loss function space is called the Pareto front.*

## 4 THREE-STAGE OPTIMIZATION APPROACH FOR EFFL

### 4.1 OPTIMIZATION PATH TO OBTAIN PARETO OPTIMAL

Fig. 1 (b) illustrates the feasible decision space of EFFL, which is bounded by the intersection of two hypothesis sets: $\mathcal{H}_B$ containing all hypothesis satisfy the $\epsilon_b$-decision bias constraint, and $\mathcal{H}_E$ containing all hypothesis satisfy the $\epsilon_{vl}$-egalitarian fairness on accuracy and $\epsilon_{vb}$-egalitarian fairness on decision bias. We aim to search the Pareto set $\mathcal{H}^*$ for the objectives in Eq. 3 within the feasible decision space. The properties of the hypothesis sets are as follows: (1) The $\mathcal{H}_B$ contains hypotheses satisfying $\epsilon_b$-decision bias in each client,

$$\{f_k(h)\}_{k=1}^N \leq \epsilon_b, \forall h \in \mathcal{H}_B. \quad (4)$$

(2) The $\mathcal{H}_E$ contains hypotheses that satisfy $\epsilon_{vl}$-egalitarian fairness on accuracy and $\epsilon_{vb}$-egalitarian fairness on decision bias across all clients,

$$\{|l_k(h) - \bar{l}(h)|\}_{k=1}^N \leq \epsilon_{vl}, \{|f_k(h) - \bar{f}(h)|\}_{k=1}^N \leq \epsilon_{vb}, \forall h \in \mathcal{H}_E. \quad (5)$$

(3) The $\mathcal{H}^* \subset \mathcal{H}_B \cap \mathcal{H}_E$ is the Pareto set of EFFL in Eq. 3, i.e.,

$$h' \nprec h, \forall h \in \mathcal{H}^*, \forall h' \in \mathcal{H}_B \cap \mathcal{H}_E. \quad (6)$$

Note that $\mathcal{H}_B \cap \mathcal{H}_E \neq \emptyset$ is not empty, at least it includes $h \in \mathcal{H}_E$ that satisfies $\bar{f}(h) \leq \epsilon_b - \epsilon_{vb}$. Since $h \in \mathcal{H}_E$, we have $\{|f_k(h) - \bar{f}(h)|\}_{k=1}^N \leq \epsilon_{vb}$, which implies that $h$ also satisfies $\{f_k(h)\}_{k=1}^N \leq \epsilon_b$ and belongs to $\mathcal{H}_B$. Finding the Pareto set for EFFL is nontrivial as the feasible decision space is highly restricted. Moreover, when the number of objectives $N$ is large, optimizing one objective may adversely affect other objectives. We construct an approximate Pareto front by linear scalarization technique. Average weights are applied to each objective and combine $N$-objectives into a single surrogate objective. The surrogate objective forms the convex part of the Pareto front, as shown in Fig. 1 (b), which is denoted as $\mathcal{H}_{\bar{L}}$. The hypothesis in the $\mathcal{H}_{\bar{L}}$ satisfies:

$$\bar{l}(h) \leq \bar{l}(h'), \forall h \in \mathcal{H}_{\bar{L}}, h' \notin \mathcal{H}_{\bar{L}}. \quad (7)$$

Compared to $\mathcal{H}^*$, $\mathcal{H}_{\bar{L}}$ is easier to obtain and can serve as an intermediate set, from which we propose a three-stage optimization algorithm with an optimal path: $h^0 \rightarrow \mathcal{H}_B \cap \mathcal{H}_{\bar{L}} \rightarrow \mathcal{H}_B \cap \mathcal{H}_E \cap \mathcal{H}_{\bar{L}} \rightarrow \mathcal{H}^*$ (purple arrows in Fig. 1 (b)), and decompose the EFFL problem into three problems as follows:

**Stage 1: Constrained Minimization Problem.** We define a constrained minimization problem on the hypothesis set $\mathcal{H}$ to obtain a hypothesis $h' \in \mathcal{H}_B \cap \mathcal{H}_{\bar{L}}$,

$$\min_{h \in \mathcal{H}} \bar{l}(h), \text{s.t.} \{f_k(h)\}_{k=1}^N \leq \epsilon_b. \quad (8)$$

By solving Eq. 8, we obtain $h'$ that 1) satisfies $\epsilon_b$-decision bias for each client and 2) minimizes the average empirical risk loss among all clients.

**Stage 2: Multi-Constrained Optimization Problem.** We formulate a multi-constrained optimization problem to obtain a hypothesis $h'' \in \mathcal{H}_B \cap \mathcal{H}_E \cap \mathcal{H}_{\bar{L}}$,

$$\min_{h \in \mathcal{H}} \bar{l}(h), \text{s.t. } \left\{ \left| l_k(h) - \bar{l}(h) \right| \right\}_{k=1}^N \leq \epsilon_{vl}, \left\{ \left| f_k(h) - \bar{f}(h) \right| \right\}_{k=1}^N \leq \epsilon_{vb}, \{ f_k(h) \}_{k=1}^N \leq \epsilon_b. \quad (9)$$

By solving Eq. 9, we obtain $h''$ that, compared to $h'$, exhibits the following properties: 1) it provides $\epsilon_{vl}$-egalitarian fairness on accuracy; and 2) it provides $\epsilon_{vb}$-egalitarian fairness on decision bias.

**Stage 3: Multi-Constrained Pareto Optimization Problem.** Focusing solely on minimizing weighted sum $\bar{l}(h)$ during optimization may harm individual clients. To address this issue, we formulate a multi-constrained Pareto optimization problem to further optimize $h''$ to $h^* \in \mathcal{H}^*$, where the empirical risk loss of each client is further reduced until Pareto optimality is achieved. At this point, the loss of each client cannot be further minimized without adversely affecting the loss of other clients,

$$\min_{h \in \mathcal{H}} \{ l_1(h), ..., l_N(h) \}, \text{s.t. } \{ f_k(h) \}_{k=1}^N \leq \epsilon_b, \left\{ \left| l_k(h) - \bar{l}(h) \right| \right\}_{k=1}^N \leq \epsilon_{vl},$$
$$\left\{ \left| f_k(h) - \bar{f}(h) \right| \right\}_{k=1}^N \leq \epsilon_{vb}, \bar{l}(h) \leq \bar{l}(h'') . \quad (10)$$

### 4.2 Three-Stage Optimization for obtaining $\mathcal{H}^*$

To obtain the convergent solution of the sub-problems defined in Eq. 8~Eq. 10, we propose a gradient-based algorithm for obtaining $h^* \in \mathcal{H}^*$, which is suitable for the implementation under FL. Given a hypothesis $h_{\theta^t}$ parameterized by $\theta^t$. At iteration $t+1$, the update rule of gradient-based methods is $\theta^{t+1} = \theta^t + \eta d$, where $d$ is a gradient descent direction and $\eta$ is the step size. In the case of an optimization problem with $N$ objectives, i.e., $\min \{ l_1(h_\theta), ..., l_N(h_\theta) \}$, a gradient $d$ is efficient to make the optimization proceed towards minimization if $\left\{ d^{*T} \nabla_\theta l_i(h_\theta) \right\}_{i=1}^N \leq 0$. As the gradient direction $d$ resides within the convex hull of the gradients of all objectives and constraints, denoted as $G = [\nabla_\theta l_1(h_\theta), ..., \nabla_\theta l_N(h_\theta)]$ (Désidéri, 2012), we can obtain a gradient descent direction $d^*$ by performing a linear transformation on $G$ using an $N$-dimensional vector $\alpha^*$,

$$d^* = \alpha^{*T} G, \text{ where } \alpha^* = \arg \min_\alpha \left\| \sum_{i=1}^N \alpha_i \nabla_\theta l_i(h_\theta) \right\|,$$
$$\text{s.t. } \sum_{i=1}^N \alpha_i = 1, \alpha_i \geq 0, \forall i \in [N] . \quad (11)$$

**Solution for Stage 1.** We first transform Eq. 8 into an equivalent single-constraint optimization problem by imposing constraint only on the max-value as follows,

$$\min_{h \in \mathcal{H}} \bar{l}(h), \text{ s.t. } \max \{ f_k(h) \}_{k=1}^N \leq \epsilon_b. \quad (12)$$

Denoting the $\max \{ f_k(h) \}_{k=1}^N$ as $f_{max}(h)$, the descent gradient of Eq. 12 lies in the convex hull of $G' = \left[ \nabla_\theta \bar{l}(h), \nabla_\theta f_{max}(h) \right]$. We employ an alternating optimization strategy: if the $\epsilon_b$-decision bias is satisfied within the worst-case client, only $\bar{l}(h)$ is further optimized,

$$d^* = \arg \min_{d \in G'} d^T \nabla_\theta \bar{l}(h), \text{ if } f_{max}(h) \leq \epsilon_b. \quad (13)$$

Otherwise, we optimize towards a descent direction $d$, which minimizes $f_{max}(h)$ while ensuring that $\bar{l}(h)$ does not increase, as follows:

$$d^* = \arg \min_{d \in G'} d^T \nabla_\theta f_{max}(h), \text{ s.t. } d^T \nabla_\theta \bar{l}(h) \leq 0, \text{ if } f_{max}(h) > \epsilon_b. \quad (14)$$

The gradient direction in Eq. 13 and Eq. 14 is optimized towards to reduce the loss while satisfying the $\epsilon_b$−decision bias constraint better, leading to a hypothesis $h'$ that balances the trade-off between loss and decision bias.

**Solution for Stage 2.** To reduce the computational complexity of handling $O(N)$-constraints in Eq. 9, we optimize the egalitarian fairness for the worst-case client. Moreover, to better achieve $\epsilon_{vl}$−egalitarian fairness on accuracy and $\epsilon_{vb}$−egalitarian fairness on decision bias, we modify Eq.

9 by treating egalitarian fairness as objectives and applying a constraint $\bar{l}(h) \leq \bar{l}(h')$ to avoid the degradation of model performance. We optimize egalitarian fairness of accuracy and decision bias alternately, i.e., if $\left\{ \left| l_k(h) - \bar{l}(h) \right| \right\}_{k=1}^{N} \leq \epsilon_{vl}$,

$$\min_{h \in \mathcal{H}} \max \left\{ \left| f_k(h) - \bar{f}(h) \right| - \epsilon_{vb} \right\}_{k=1}^{N}, \text{ s.t. } \max \left\{ f_k(h) \right\}_{k=1}^{N} \leq \epsilon_b, \bar{l}(h) \leq \bar{l}(h'), \quad (15)$$

else,

$$\min_{h \in \mathcal{H}} \max \left\{ \left| l_k(h) - \bar{l}(h) \right| - \epsilon_{vl} \right\}_{k=1}^{N},$$
$$\text{s.t. } \max \left\{ \left| f_k(h) - \bar{f}(h) \right| \right\}_{k=1}^{N} \leq \epsilon_{vb}, \max \left\{ f_k(h) \right\}_{k=1}^{N} \leq \epsilon_b, \bar{l}(h) \leq \bar{l}(h'). \quad (16)$$

Denoting the $\max \left\{ \left| l_k(h) - \bar{l}(h) \right| - \epsilon_{vl} \right\}_{k=1}^{N}$ and $\max \left\{ \left| f_k(h) - \bar{f}(h) \right| - \epsilon_{vb} \right\}_{k=1}^{N}$ as $\hat{l}_{max}(h)$ and $\hat{f}_{max}(h)$, respectively, the gradient descent direction of Eq. 15 lies in the convex hull of $G'' = \left[ \nabla_\theta \hat{f}_{max}, \nabla_\theta f_{max}, \nabla_\theta \bar{l} \right]$. We obtain the optimal $d^*$ as follows:

$$d^* = \arg \min_{d \in G''} d^T \nabla_\theta \hat{f}_{max}(h), \text{s.t. } d^T \nabla_\theta \bar{l}(h) \leq 0, d^T \nabla_\theta f_{max}(h) \leq 0 \text{ if } f_{max}(h) > \epsilon_b. \quad (17)$$

The gradient descent direction of Eq. 16 lies in the convex hull of $G'' = \left[ \nabla_\theta \hat{l}_{max}, \nabla_\theta \hat{f}_{max}, \nabla_\theta f_{max}, \nabla_\theta \bar{l} \right]$. We obtain the optimal $d^*$ as follows:

$$d^* = \arg \min_{d \in G''} d^T \nabla_\theta \hat{l}_{max}(h), \text{s.t. } d^T \nabla_\theta \bar{l}(h) \leq 0,$$
$$d^T \nabla_\theta \hat{f}_{max}(h) \leq 0 \text{ if } \hat{f}_{max}(h) > \epsilon_{vb}, \ d^T \nabla_\theta f_{max}(h) \leq 0 \text{ if } f_{max}(h) > \epsilon_b. \quad (18)$$

The constraints are dynamically imposed, depending on whether the current hypotheses $h$ satisfies $\epsilon_b$-decision bias and $\epsilon_{vb}$-egalitarian fairness on the decision bias. The optimal gradient direction in Eq. 17 and Eq. 18 is optimized towards improving the equality of performance among clients without causing the degradation of model performance, leading to a hypothesis $h''$ that balances the trade-off between egalitarian fairness and maximizing model performance.

**Solution for Stage 3.** To reduce the computational complexity of minimizing $N$ objectives and handling $O(N)$-constraints in Eq. 10, we optimize the empirical risk loss for the worst-case client and impose constraints $l_k(h) \leq l_k(h'')$ to prevent the degradation of performance for other clients.

$$\min_{h \in \mathcal{H}} \max \left\{ l_1(h), ..., l_N(h) \right\}, \text{s.t. } l_k(h) \leq l_k(h''), \forall k \in [N], \bar{l}(h) \leq \bar{l}(h''),$$
$$\max \left\{ f_k(h) \right\}_{k=1}^{N} \leq \epsilon_b, \max \left\{ \left| l_k(h) - \bar{l}(h) \right| \right\}_{k=1}^{N} \leq \epsilon_{vl}, \max \left\{ \left| f_k(h) - \bar{f}(h) \right| \right\}_{k=1}^{N} \leq \epsilon_{vb}. \quad (19)$$

Denoting the $\max \left\{ l_k(h) \right\}_{k=1}^{N}$ as $l_{max}(h)$, the gradient descent direction lies in the convex hull of $G^* = [\nabla_\theta l_{max}(h), \nabla_\theta l_1(h), ..., \nabla_\theta l_N(h), \nabla_\theta \bar{l}(h), \nabla_\theta f_{max}(h), \nabla_\theta \hat{l}_{max}(h), \nabla_\theta \hat{f}_{max}(h)]$. We obtain the optimal $d^*$ as follows,

$$d^* = \arg \min_{d \in G^*} d^T \nabla_\theta l_{max}(h),$$
$$\text{s.t. } d^T \nabla_\theta \bar{l}(h) \leq 0, \ d^T \nabla_\theta f_{max}(h) \leq 0 \text{ if } f_{max}(h) > \epsilon_b,$$
$$d^T \nabla_\theta \hat{l}_{max}(h) \leq 0 \text{ if } \hat{l}_{max}(h) > \epsilon_{vl}, d^T \nabla_\theta \hat{f}_{max}(h) \leq 0 \text{ if } \hat{f}_{max}(h) > \epsilon_{vb},$$
$$d^T \nabla_\theta l_i(h) \leq 0, \forall i \in [N] \text{ and } i \neq \arg \max \left\{ l_k(h) \right\}_{k=1}^{N}. \quad (20)$$

The constraints are dynamically imposed, depending on whether the current hypotheses $h$ satisfies $\epsilon_b$-decision bias, $\epsilon_{vl}$-egalitarian fairness on the accuracy and $\epsilon_{vb}$-egalitarian fairness on the decision bias. The optimal gradient direction in Eq. 20 is optimized to minimize the empirical risk loss on the worst client without deteriorating the performance of other clients, resulting in a Pareto optimal hypothesis $h^*$. The algorithm implementation in FL is described in the Appx. A.

## 5 EXPERIMENTS

### 5.1 SETTINGS

**Datasets.** ❶ synthetic dataset with 2 clients, ❷ Adult (Kohavi & Becker, 1996): real-world dataset with 2 clients, ❸ eICU (Johnson et al., 2018): real-world dataset with 11 clients.

Table 1: The test performance on three datasets.

| Dataset | Method | Model Performance | | | |
|---|---|---|---|---|---|
| | | Local Acc. | | Local Bias | |
| | | Avg. | Std.($\epsilon_{vl}$) | Avg.($\epsilon_b$) | Std.($\epsilon_{vb}$) |
| Synthetic $\epsilon_b = 0.1$ $\epsilon_{vl} = 0.01$ $\epsilon_{vb} = 0.04$ | FedAvg | **.7735** | .0283($\times$) | .2480($\times$) | .0819($\times$) |
| | q-FFL | **.7735** | .0283($\times$) | .2480($\times$) | .0819($\times$) |
| | Ditto | .7229 | .0132($\times$) | .2703($\times$) | .0566($\times$) |
| | FedMDFG | .7717 | **.0068**($\checkmark$) | .2473($\times$) | .0662($\times$) |
| | FedAvg+FairBatch | .6360 | .0643($\times$) | .1040($\approx$) | .0798($\times$) |
| | FedAvg+FairReg | .6227 | .0394($\times$) | .0952($\checkmark$) | .0463($\times$) |
| | FCFL | .6330 | .0177($\times$) | .0812($\checkmark$) | .0435($\approx$) |
| | EFFL | .6327 | .0087($\checkmark$) | **.0801**($\checkmark$) | **.0359**($\checkmark$) |
| Adult $\epsilon_b = 0.01$ $\epsilon_{vl} = 0.03$ $\epsilon_{vb} = 0.005$ | FedAvg | **.7767** | .0592($\times$) | .0328($\times$) | .0093 ($\times$) |
| | q-FFL | .7662 | .0400($\times$) | .0472($\times$) | .0046($\checkmark$) |
| | Ditto | .7210 | **.0039**($\checkmark$) | .0169($\times$) | .0111($\times$) |
| | FedMDFG | .7629 | .0397($\times$) | .0436($\times$) | .0068($\approx$) |
| | FedAvg+FairBatch | .7756 | .0556($\times$) | .0136($\times$) | .0128($\times$) |
| | FedAvg+FairReg | .7663 | .0686($\times$) | .0089($\checkmark$) | .0066($\approx$) |
| | FCFL | .7638 | .0487($\times$) | .0143($\times$) | .0159($\times$) |
| | EFFL | .7685 | .0281($\checkmark$) | **.0036**($\checkmark$) | **.0009**($\checkmark$) |
| eICU $\epsilon_b = 0.02$ $\epsilon_{vl} = 0.02$ $\epsilon_{vb} = 0.02$ | FedAvg | .6560 | .0427($\times$) | .0371($\times$) | .0409($\times$) |
| | q-FFL | **.6565** | .0425($\times$) | .0371($\times$) | .0405($\times$) |
| | Ditto | .6311 | .0216($\approx$) | .0472($\times$) | .0447($\times$) |
| | FedMDFG | .6479 | .0227($\times$) | .0311($\times$) | .0266($\times$) |
| | FedAvg+FairBatch | .6441 | .0413($\times$) | .0304($\times$) | .0298($\times$) |
| | FedAvg+FairReg | .6455 | .0408($\times$) | .0322($\times$) | .0266($\times$) |
| | FCFL | .6550 | .0272($\times$) | .0344($\times$) | .0246($\times$) |
| | EFFL | .6530 | **.0195**($\checkmark$) | **.0209**($\approx$) | **.0201**($\approx$) |

**xxx** : Best performance compared to all algorithms.
($\times$) : Violation of constraint exceeds 10%.
($\approx$) : Close to constraint, with violation of constraint not exceeding 10%.
($\checkmark$) : Satisfy constraint.

**Baselines.** ❶ FedAvg (McMahan et al., 2017), ❷ FedAvg + FairBatch (Roh et al., 2021), ❸ FedAvG+FairReg, ❹ Ditto (Li et al., 2021), ❺ q-FFL (Li et al., 2019), ❻ FCFL (Cui et al., 2021), ❼ FedMDFG (Pan et al., 2023).

**Hyperparameters.** We divide the communication rounds into three stages, each with $750$, $750$, and $500$ rounds, respectively, to ensure that the global model is fully updated and converges in each stage. In the constraint budgets setting, we set the decision bias budget $\epsilon_b$, the egalitarian fairness budget on accuracy $\epsilon_{vl}$, and the egalitarian fairness budget on decision bias $\epsilon_{vb}$ to half of the related-performance achieved by the original FedAvg. For example, as shown in Tab. 1, on the synthetic dataset experiments, the decision bias Avg. of FedAvg is $0.2480$, so we set $\epsilon_b = 0.1$. The accuracy Std. of FedAvg is $0.0283$, so we set $\epsilon_{vl} = 0.01$. The decision bias Std. of FedAvg is $0.0819$, so we set $\epsilon_{vb} = 0.04$. We use the same parameter-setting strategy for other datasets. Since the constraints may conflict with each other, this setting allows us to better evaluate the superior performance of our proposed EFFL method and avoid making a constraint too tight, which may result in a solution that is only optimal on this constraint.

**Evaluation Metrics.** To evaluate the effectiveness and equality of the global model's performance across all clients, we introduce three evaluation metrics under the accuracy and decision bias performance, respectively: ❶ Avg.: the average performance of the global model across all clients; ❷ Std.: the variation of the performance of the global model across clients. We utilize the TPSD as metric for decision bias.

**More details about the settings are in the Appx. B.**

## 5.2 ACCURACY, DECISION BIAS AND EGALITARIAN FAIRNESS

We compare the global model performance of our proposed EFFL with other SOTA baselines on three datasets. In the EFFL problem setting, we introduce three types of constraints $\epsilon_b-$decision bias, $\epsilon_{vl}-$egalitarian fairness on accuracy and $\epsilon_{vb}-$egalitarian fairness on decision bias. As shown in Tab. 1, our method achieves the best satisfaction of the three constraints, with strict satisfaction under Synthetic and Adult datasets, and approximate satisfaction under the eICU dataset. The SOTA baselines are not able to guarantee all three constraints simultaneously. In terms of accu-

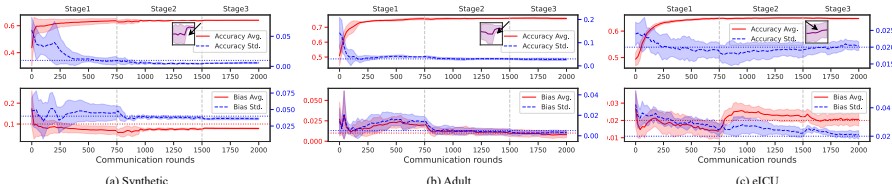

Figure 2: Testing results during 2000 communication rounds.

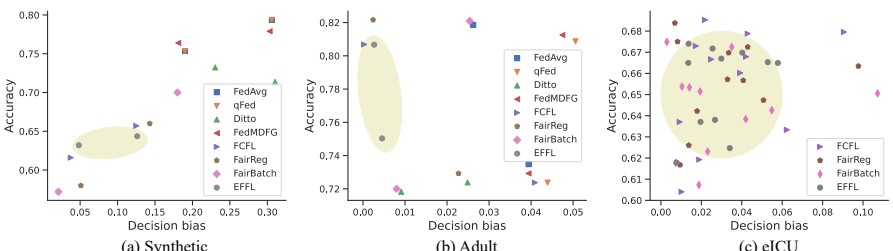

Figure 3: The distribution of model performance across different clients.

racy, the methods that do not consider decision bias (FedAvg, q-FFL, Ditto, and FedMDFG) have higher accuracy. However, as there is a trade-off between accuracy and decision bias, we compare our method with the baselines that also aim to reduce decision bias (FedAvg+FairBatch, FedAvg+FairReg, FCFL). Our method achieves the best constraint satisfaction with only $0.3\%$ decrease in accuracy on the synthetic dataset, $0.7\%$ decrease in accuracy on the adult dataset, and $0.2\%$ decrease in accuracy on the eICU dataset.

Fig. 2 illustrates the convergent efficiency of EFFL within 2000 communication rounds on three datasets. The experiments are repeated 20 times. We observe that the accuracy of the global model converges as the communication rounds increase, while the decision bias and the egalitarian fairness of accuracy and decision bias remain within the predefined budgets (indicated by the colored dashed lines in Fig. 2). We provide the theoretical proof of convergence in Appx. C.

Fig. 3 illustrates the distribution of model performance across various clients. We conduct experiments on three datasets. For a clearer visualization, we prioritize the display of baselines considering decision bias (FedAvg+Fairbatch, FedAvg+FairReg, FCFL) on the eICU dataset, which encompasses 11 clients. The results demonstrate that our proposed EFFL model ensures a more equitable performance distribution among clients, thereby indicating enhanced egalitarian fairness.

In EFFL, training objectives and constraints are imposed on each client individually in order to consider local performance and egalitarian fairness. This ensures that our method maintains scalability even when dealing with a larger number of clients. **Additional experiments on APSD bias metric, scalability, robustness, hyperparameter settings, and ablation studies are provided in Appx. D**.

## 6 CONCLUSION

In this paper, we have investigated the egalitarian fairness issues in federated learning (FL), which have significant impacts on the sustainability of the FL system due to the **Matthew Effect**. We have analyzed the possible trade-offs for achieving egalitarian fairness and have formally defined **Egalitarian Fairness Federated Learning** (EFFL) as a multi-constrained multi-objectives optimization problem. Furthermore, we have designed an effective optimization path that decomposed the original problem into three sub-problems and proposed a three-stage algorithm to achieve Pareto optimal solutions under trade-offs. In the end, we have conducted a thorough empirical evaluation to demonstrate that our proposed method outperforms other state-of-the-art baselines in achieving a high-performance global model with enhanced egalitarian fairness among all clients.

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

## A  ALGORITHM IMPLEMENT IN FEDERATED LEARNING

We give the implementation details of our proposed EFFL in Alg. 1:

## B  EXPERIMENTAL IMPLEMENTATION DETAILS

### B.1  DATASETS DETAILS

We adopt the following datasets, including:

(1) Synthetic dataset: we generate a synthetic dataset with a protected attribute: $A \sim Ber(0.5)$, two general attributes: $X_1 \sim \mathcal{N}(0, 1)$, $X_2 \sim \mathcal{N}(\mathbb{1}(a > 0), 2)$. The label is setting by $Y \sim Ber(u^l \mathbb{1}(x_1 + x_2 \leq 0) + u^h \mathbb{1}(x_1 + x_2 > 0))$, where $\{u^l, u_h\} = \{0.3, 0.6\}$ if $A = 0$, otherwise, $\{u^l, u^h\} = \{0.1, 0.9\}$. We split the dataset into two clients based on whether $x_1 \leq -0.5$ to make the clients heterogeneous in distribution and size;

(2) Adult dataset: Adult is a binary classification dataset with more than $40000$ adult records for predicting whether the annual income is greater than $50K$ (Kohavi & Becker, 1996). We split the dataset into two clients based on whether the individual's education level is a Ph.D and select *race* as a protected attribute;

(3)EICU dataset: the eICU dataset includes data records about clinical information and hospital details of patients who are admitted to ICUs. The dataset has been processed following the steps outlined in (Johnson et al., 2018). We filter out the hospitals with data points less than $1500$, leaving 11 hospitals in our experiments. Naturally, we treat each hospital as a client.

---

**Algorithm 1** EFFL Implementation.

---

1: **Input:** Communication rounds for three stages: $T^1$, $T^2$ and $T^3$; the decision bias budget $\epsilon_b$ and the egalitarian fairness budgets: $\epsilon_{vl}$ and $\epsilon_{vb}$.
2: Total communication rounds: $T \leftarrow T^1 + T^2 + T^3$;
3: **for** $t \in [0, T]$ **do**
4:    The $k$-th client receives global model $h^t$ from the server, where $k \in \{1, ..., N\}$;
5:    The $k$-th client evaluates its local performance by the loss function $l_k(h^t)$ and the decision bias function $f_k(h^t)$, where $k \in \{1, ..., N\}$;
6:    The $k$-th client computes the gradients of local loss and local decision bias, $\nabla_\theta l_k(h^t)$ and $\nabla_\theta f_k(h^t)$, where $k \in \{1, ..., N\}$;
7:    Clients send performance evaluations and gradients to the server;
8:    Server calculates:

$$\nabla_\theta \bar{l}(h) \leftarrow \frac{1}{N} \sum_{k=1}^{N} \nabla_\theta l_k(h); \tag{21}$$

9:    **if** $t \in \left[0, T^1\right]$ **then**
10:       The server determines the optimal gradient descent direction $d^*$ by solving:

$$\begin{cases} d^* = \arg \min_{d \in G'} d^T \nabla_\theta \bar{l}(h), \text{if } f_{max}(h) \leq \epsilon_b, \\ d^* = \arg \min_{d \in G'} d^T \nabla_\theta f_{max}(h), \text{s.t. } d^T \nabla_\theta \bar{l}(h) \leq 0, \text{else.} \end{cases} \tag{22}$$

11:    **end if**
12:    **if** $t \in \left[T^1, T^1 + T^2\right]$ **then**
13:       The server determines the optimal gradient descent direction $d^*$ by solving:

$$\begin{cases} d^* = \arg \min_{d \in G''} d^T \nabla_\theta \hat{f}_{max}(h), \\ \text{s.t. } d^T \nabla_\theta \bar{l}(h) \leq 0, d^T \nabla_\theta f_{max}(h) \leq 0 \text{ if } f_{max}(h) > \epsilon_b. & \text{if } \left\{\left|l_k(h) - \bar{l}(h)\right|\right\}_{k=1}^{N} \leq \epsilon_{vl}, \\ d^* = \arg \min_{d \in G''} d^T \nabla_\theta \hat{l}_{max}(h), \text{s.t. } d^T \nabla_\theta \bar{l}(h) \leq 0, \\ d^T \nabla_\theta \hat{f}_{max}(h) \leq 0 \text{ if } \hat{f}_{max}(h) > \epsilon_{vb}, \ d^T \nabla_\theta f_{max}(h) \leq 0 \text{ if } f_{max}(h) > \epsilon_b. & \text{, else.} \end{cases} \tag{23}$$

14:    **end if**
15:    **if** $t \in \left[T^1 + T^2, T^1 + T^2 + T^3\right]$ **then**
16:       The server determines the optimal gradient descent direction $d^*$ by solving;

$$\begin{aligned} d^* = \arg \min_{d \in G^*} d^T \nabla_\theta l_{max}(h), \text{s.t. } d^T \nabla_\theta \bar{l}(h) \leq 0, \ d^T \nabla_\theta f_{max}(h) \leq 0 \text{ if } f_{max}(h) > \epsilon_b, \\ d^T \nabla_\theta \hat{l}_{max}(h) \leq 0 \text{ if } \hat{l}_{max}(h) > \epsilon_{vl}, d^T \nabla_\theta \hat{f}_{max}(h) \leq 0 \text{ if } \hat{f}_{max}(h) > \epsilon_{vb}, \\ d^T \nabla_\theta l_i(h) \leq 0, \forall i \in [N] \text{ and } i \neq \arg \max \{l_k(h)\}_{k=1}^{N}. \end{aligned} \tag{24}$$

17:    **end if**
18:    Update the parameter of model $h^{t+1}$: $\theta^{t+1} \leftarrow \theta^t + \eta d^*$;
19: **end for**
20: **return:** Pareto optimal hypothesis in feasible decision space: $h^T$.

---

## B.2 BASELINES DETAILS

We adopt the following baselines, including:

(1) FedAvg (McMahan et al., 2017): the original FL algorithm for distributed training of private data. It does not consider fairness for different demographic groups and different clients;

(2) FedAvg + FairBatch (Roh et al., 2021): each client adopts the state-of-the-art FairBatch in-processing debiasing strategy on its local training data and then aggregation uses FedAvg;

(3) FedAvG+FairReg: a local processing method by optimizing the linear scalarized objective with the fairness regularizations of all clients;

(4) Ditto (Li et al., 2021): an FL framework to achieve training fairness by allowing clients to run finetuning the received global model on the local data;

(5) q-FFL (Li et al., 2019): an FL framework to achieve fair loss among clients by weighing different local clients differently by taking the q-th power of the local empirical loss;

(6) FCFL (Cui et al., 2021): an FL framework to achieve algorithmic fairness across different local clients and consistency by minimizing the model loss of the worst-performed client;

(7) FedMDFG (Pan et al., 2023): a multiple gradient descent algorithm (MGDA)-based method by adding cosine similarity between the loss vectors among clients and the unit vector as a fairness objective.

### B.3 NORMALIZATION

We suggest normalizing the gradients before computing the optimal gradient descent direction $d^*$ when there is a large disparity among the gradients, as in the eICU-related experiments where the number of clients is large. We experiment with three different methods of gradient normalization and find that the $L_2$-based method performs the best in practice. The gradient $g = \nabla_\theta l(h_\theta)$ can be normalized by the following three methods:

$$L_2 - \text{based} : \mathbf{g} = \mathbf{g}/\sqrt{\sum_{i=1}^{|g|} g_i^2}. \tag{25}$$

$$\text{Loss} - \text{based} : \mathbf{g} = \mathbf{g}/l(h_\theta). \tag{26}$$

$$L_2 + \text{Loss} - \text{based} : \mathbf{g} = \mathbf{g}/\left(\sqrt{\sum_{i=1}^{|g|} g_i^2} \times l(h_\theta)\right). \tag{27}$$

## C  ADDITIONAL ANALYSIS

### C.1  TIME COMPLEXITY ANALYSIS

Ignoring the communication time between clients and server, in a scenario with $N$ clients, the time consumption involved in solving EFFL given by Alg. 1 mainly comes from: (1) the gradient computation on local clients, which is $O(|\mathcal{D}_k| d)$ for $k$-th client, where $|\mathcal{D}_k|$ is the size of the dataset $\mathcal{D}_k$ owned by the $k$-th client and $d$ is the data feature dimension; (2) the $d^*$ solving: for a linear programming problem, $\min_{x \in \mathbb{R}^n} A^T x$, s.t. $Cx = b, x \geq 0$ with $n$ variables and $\Omega(n)$ constraints, it can be solved in polynomial time of $\widetilde{O}^*(n^\omega)$ (Alman & Williams, 2021), where $\omega$ is the exponent of matrix multiplication. The current best known upper bound is $\omega < 2.3728596$ (Cohen et al., 2021), which implies a time complexity not exceeding $\widetilde{O}^*(n^{2.38})$. Therefore, in our proposed three-stage gradient descent direction $d^*$ solving, the time consumption does not exceed $\widetilde{O}^*(2^{2.38}) + \widetilde{O}^*(4^{2.38}) + \widetilde{O}^*((N+4)^{2.38})$. Based on the above analysis, the time consumption in Stage 1 and Stage 2 is mainly affected by the gradient computation on local clients and with only a constant computation time spent on $d^*$. In Stage 3, compared to Stage 1 and Stage 2, the computation time spent on $d^*$ increases, depending on the number of clients, but overall, it is still bounded by polynomial time complexity.

### C.2  CONVERGENCE ANALYSIS

**The proposed three-stage algorithm is convergent**. We first give the following proposition:
**Proposition 1.** For $N$ optimization objectives $\{l_1(\theta^t), ..., l_N(\theta^t)\}$ and the following model parameter updating rule under a gradient direction $d$: $\theta^{t+1} = \theta^t + \eta d$; if $d^T \nabla_\theta l_i \leq 0$, there exists $\eta_0$ such that for $\forall \eta \in [0, \eta_0]$, the objectives $\{l_1(\theta^t), ..., l_N(\theta^t)\}$ will not increase (decrease or remain unchanged), that is, the iterations toward convergent.

**Proof.** The above proposition can be proved by performing a first-order Taylor expansion at $\theta^t$, i.e., $l_i(\theta^{t+1}) = l_i(\theta^t) + \nabla_\theta l_i(\theta^t)(\theta^t + \eta d - \theta^t) + R_1(\theta^t + \eta d)$, where $R_1$ is the first order remainder term, which is a higher-order infinitesimal of $\eta$, denoted by $o(\eta)$ (according to the Taylor formula, if the function $l_i$ is second-order differentiable at $\theta^t$, then the first-order remainder term can be expressed as: $R_1(\theta) = \frac{\nabla_\theta^2 l(\theta^t)}{2!}(\theta - \theta^t)^2 \to R_1(\theta^t + \eta d) = \frac{\nabla_\theta^2 l(\theta^t)}{2!}\eta^2 d^2 = o(\eta)$). Therefore, we have $l_i(\theta^{t+1}) - l_i(\theta^t) = \eta d^T \nabla_\theta l_i(\theta^t) + o(\eta)$, since $o(\eta)$ approaches 0 faster than $\eta$, when $d^T \nabla_\theta l_i(\theta^t) \leq 0$, there exists $\eta_0 > 0$ such that for $\forall \eta \in [0, \eta_0]$, $l_i(\theta^{t+1}) - l_i(\theta^t) \leq 0$.

Based on the above proposition, the parameter update at each stage in the proposed three-stage optimization algorithm is towards a gradient descent direction $d^*$ that satisfies $d^T \nabla_\theta g_k(h) \leq 0$, where $g_k$ is the different optimization objective at different stages, as detailed in Alg. 1. Therefore, the proposed algorithm is convergent.

Table 2: The test performance on three datasets.

| Dataset | Method | Model Performance | | | |
| | | Local Acc. | | Local Bias | |
| | | Avg. | Std.($\epsilon_{vl}$) | Avg.($\epsilon_b$) | Std.($\epsilon_{vb}$) |
|---|---|---|---|---|---|
| Synthetic $\epsilon_b = 0.08$ $\epsilon_{vl} = 0.01$ $\epsilon_{vb} = 0.04$ | FedAvg | **.7735** | .0283($\times$) | .1663($\times$) | .0791($\times$) |
| | q-FFL | **.7735** | .0283($\times$) | .1663($\times$) | .0791($\times$) |
| | Ditto | .7229 | .0132($\times$) | .1977($\times$) | .0615($\times$) |
| | FedMDFG | .7717 | .0068($\checkmark$) | .1667($\times$) | .0718($\times$) |
| | FedAvg+FairBatch | .6695 | .0397($\times$) | .0999($\times$) | **.0401**($\checkmark$) |
| | FedAvg+FairReg | .6191 | .0250($\times$) | .0976($\times$) | .0720($\times$) |
| | FCFL | .6302 | .0165($\times$) | .0687($\checkmark$) | 0453($\times$) |
| | EFFL | .6269 | **.0029**($\checkmark$) | **.0621**($\checkmark$) | .0430($\approx$) |
| Adult $\epsilon_b = 0.02$ $\epsilon_{vl} = 0.03$ $\epsilon_{vb} = 0.01$ | FedAvg | **.7767** | .0592($\times$) | .0494($\times$) | .0257($\times$) |
| | q-FFL | .7662 | .0400($\times$) | .0386($\times$) | .0089($\checkmark$) |
| | Ditto | .7158 | **.0123**($\checkmark$) | .0450 ($\times$) | .0481 ($\times$) |
| | FedMDFG | .7656 | .0397($\times$) | .0436($\times$) | .0068($\checkmark$) |
| | FedAvg+FairBatch | .7726 | .0556 ($\times$) | .0218($\approx$) | .0076 ($\checkmark$) |
| | FedAvg+FairReg | .7446 | .0484($\times$) | .0109($\checkmark$) | .0186($\times$) |
| | FCFL | .7583 | .0487($\times$) | .0109($\checkmark$) | .0195($\times$) |
| | EFFL | .7549 | .0284($\checkmark$) | **.0101**($\checkmark$) | **.0067**($\checkmark$) |
| eICU $\epsilon_b = 0.02$ $\epsilon_{vl} = 0.02$ $\epsilon_{vb} = 0.02$ | FedAvg | .6460 | .0427($\times$) | .0386($\times$) | .0310($\times$) |
| | q-FFL | .6465 | .0425($\times$) | .0384($\times$) | .0307($\times$) |
| | Ditto | .6305 | .0216($\approx$) | .0399($\times$) | .0405($\times$) |
| | FedMDFG | .6497 | .0217($\approx$) | .0395($\times$) | .0301($\times$) |
| | FedAvg+FairBatch | .6441 | .0213($\approx$) | .0301($\times$) | .0227($\times$) |
| | FedAvg+FairReg | .5346 | .0765($\times$) | .0335($\times$) | .0267($\times$) |
| | FCFL | **.6551** | .0362($\times$) | .0331($\times$) | .0200($\checkmark$) |
| | EFFL | .6531 | **.0192**($\checkmark$) | **.0207**($\approx$) | **.0182**($\checkmark$) |

**xxx** : Best performance compared to all algorithms.
($\times$) : Violation of constraint exceeds 10%.
($\approx$) : Close to constraint, with violation of constraint not exceeding 10%.
($\checkmark$) : Satisfy constraint.

## C.3 STABILITY ANALYSIS

Given gradients from the clients have an error $\tilde{G}(\theta^t) = G(\theta^t) + \mathbf{e}^t$, then $\|\tilde{\alpha} - \alpha\|_2 \leq \mathcal{O}(max_t \|e^t\|_2)$. Following that, the error of the model parameter $\theta^{t+1}$ is bounded by:

$$
\begin{aligned}
\left\|\tilde{\theta}^{t+1} - \theta^{t+1}\right\|_2 &= \eta \left\|\tilde{\alpha}^T G + \tilde{\alpha}^T e^t - \alpha^T G\right\|_2 \\
&\leq \eta \|G\|_2 \mathcal{O}\left(\max_t \|e^t\|_2\right) + \eta \mathcal{O}\left(\max_t \|e^t\|_2\right)\|e^t\|_2 + \|a\|_2 \|e^t\|_2
\end{aligned}
\tag{28}
$$

Given that $\eta$, $\|G\|_2$, and $\|a\|_2$ are bounded, $\left\|\tilde{\theta}^{t+1} - \theta^{t+1}\right\|_2 \leq \mathcal{O}(\max_t \|e^t\|_2)$ is also bounded in our algorithm, the model's sensitivity to input errors satisfies $\mathcal{O}(max_t \|e^t\|_2)$-stability.

## D ADDITIONAL EXPERIMENTS

### D.1 SUPPLEMENTARY BIAS METRICS

In this experiment, we adopt APSD as a decision bias metric and compare the global model performance of our proposed EFFL with other SOTA baselines on three datasets. As shown in Tab. 2, our method achieves the best satisfaction of the three constraints, while current SOTA baselines cannot guarantee all three constraints simultaneously. As there is a trade-off between accuracy and decision bias, we evaluate the accuracy performance by comparing with the baselines that satisfy the $\epsilon_b-$decision bias constraint. Our proposed EFFL achieves the best constraint satisfaction with only $0.3\%$ decrease in accuracy under synthetic and adult datasets and $0.2\%$ decrease in accuracy under the eICU dataset.

### D.2 SCALABILITY

We conduct experiments under the ACSPublicCoverage dataset, which is used to predict whether an individual is covered by public health insurance (Ding et al., 2021) in the United States. The

Table 3: The test performance under ACSPublicCoverage dataset with 51 clients.

| Method | Local Acc. | | Local Bias | |
|---|---|---|---|---|
| | Avg. | Std. | Avg. | Std. |
| FedAvg | 0.5778 | 0.0352 | 0.0260 | 0.0242 |
| q-FFL | 0.5888 | 0.0406 | 0.0236 | 0.0202 |
| Ditto | **0.6578** | 0.0584 | 0.0307 ↑ | 0.0328 |
| FedMDFG | 0.5978 | 0.0453 | 0.0215 | 0.0187 |
| FedAvg+FairBatch | 0.5972 | 0.0454 | 0.0214 | 0.0188 |
| FedAvg+FairReg | 0.5964 | 0.0453 | 0.0202 | 0.0194 |
| FCFL | 0.6085 | 0.0453 | 0.0215 | 0.0187 |
| EFFL | 0.6037 | **0.0284** | **0.0147** | **0.0128** |

Table 4: The test performance under attacks.

| Attack | Avg. of Local Acc.$\pm$ Std. of Local Acc./ Avg. of Local Bias$\pm$ Std. of Local Bias | | |
|---|---|---|---|
| | **Enlarge** | **Random** | **Zero** |
| FedAvg | .5307$\pm$.0414/.0205$\pm$.0669 | .5129$\pm$.0743/.0242$\pm$.0291 | .6225$\pm$.0503/.0245$\pm$.0411 |
| q-FFL | .6278$\pm$.0261/.0242$\pm$.0250 | .5824$\pm$.0461/.0225$\pm$.0371 | .6492$\pm$.0386/.0216$\pm$.0379 |
| Ditto | .6162$\pm$.0283/.0203$\pm$.0247 | .5130$\pm$.0771/.0273$\pm$.0331 | .6236$\pm$.0498/.0231$\pm$.0399 |
| FedMDFG | .6512$\pm$.0345/.0217$\pm$.0340 | .6502$\pm$.0332/.0229$\pm$.0318 | .6466$\pm$.0271/.0213$\pm$.0211 |
| FedAvg+FairBatch | .6079$\pm$.0280/.0199$\pm$.0200 | .5095$\pm$.0724/.0297$\pm$.0335 | .6225$\pm$.0503/.0245$\pm$.0411 |
| FedAvg+FairReg | .5643$\pm$.0582/.0272$\pm$.0372 | .6510$\pm$.0299/.0225$\pm$.0275 | .6511$\pm$.0292/.0228$\pm$.0283 |
| FCFL | .6280$\pm$.0296/.0232$\pm$.0237 | **.6535**$\pm$.0263/.0225$\pm$.0233 | .6254$\pm$.0317/.0215$\pm$.0295 |
| EFFL | **.6536$\pm$.0278/.0178$\pm$.0185** | .6510$\pm$**.0244/.0203$\pm$.0158** | .6522$\pm$**.0259/.0206$\pm$.0183** |

**xxx** : Best performance compared to all algorithms.

dataset is collected in the year 2022. Naturally, we treat each state as a client, generating 51 clients in our experiments. The results are shown in Tab. 3. Our method achieves superior performance in reducing decision bias and ensuring egalitarian fairness among clients. Ditto has the best accuracy but at the cost of high decision bias and significant performance disparity among clients, which may exacerbate the Matthew effect and be undesirable from a social welfare perspective.

## D.3   ROBUSTNESS

We conduct robustness validation experiments on the eICU dataset and randomly select 4 clients to be malicious. The malicious clients adopt the following attacks to dominate or disrupt the FL process: (1) Enlarge: the malicious clients enlarge the local gradients or local model parameters sent to the server to enhance their influence in the training process. In our experiments, we set the enlarging factor to 10; (2) Random: the malicious clients send randomly generated local gradients or model parameters to the server to disrupt the training process; (3) Zero: The malicious clients send zero local gradients or zero model parameters to the server to disrupt the training process. Tab. 4 shows the testing performance of the SOTA baselines and our EFFL under attacks. The performance metrics include the mean and standard deviation of accuracy and decision bias of the global model on all clients. The results demonstrate that our EFFL is robust and the performance of the global model does not degrade due to the presence of malicious clients, while other baselines suffer from different degrees of decline in testing performance. The three-stage algorithm to solve EFFL provides equal gradient descent directions that ensure that even if malicious clients are attempting to dominate the model training, EFFL can still protect the performance of the honest clients and prevent them from being discriminated, and ensures that the model's performance is not compromised under attack.

## D.4   CONSTRAINT BUDGETS

In EFFL, we introduce three constraint budgets, $\epsilon_b$, $\epsilon_{vl}$ and $\epsilon_{vb}$, to control the decision bias performance of the global model on the clients, the egalitarian fairness of the accuracy and decision bias performance distribution among the clients, respectively. To explore the effect of different budgets

Table 5: Effects of budgets $\epsilon_b$, $\epsilon_{vl}$ and $\epsilon_{vb}$.

| $\epsilon_b$ | | 0.01 | 0.02 | 0.05 | 0.1 |
|---|---|---|---|---|---|
| Avg. Acc. | Adult | 0.7549 | 0.7563 | **0.7634** | **0.7634** |
| | eICU | 0.6277 | 0.6530 | 0.6549 | **0.6590** |
| Avg. Bias | Adult | **0.0107** | 0.0118 | 0.0350 | 0.0354 |
| | eICU | **0.0136** | 0.0209 | 0.0304 | 0.0416 |
| $\epsilon_{vl}$ | | 0.01 | 0.02 | 0.05 | 0.1 |
| Std. Acc. | Adult | **0.0176** | 0.0226 | 0.0326 | 0.0327 |
| | eICU | **0.0143** | 0.0195 | 0.0239 | 0.0254 |
| $\epsilon_{vb}$ | | 0.01 | 0.02 | 0.05 | 0.1 |
| Std. Bias | Adult | **0.0087** | 0.0103 | 0.0152 | 0.0158 |
| | eICU | **0.0112** | 0.0201 | 0.0321 | 0.0361 |

Table 6: Ablation study of each stage in EFFL.

| Dataset | Stage | Local Bias | Local Acc. |
|---|---|---|---|
| Synthetic | Stage 1 | 0.6519±0.0169 | 0.1033±0.0477 |
| | Stage 2 | 0.5740±0.0661 | 0.0772±0.0740 |
| | Stage 3 | **0.6840**±0.0983 | 0.2483±0.0670 |
| | Stage 1+Stage 2+Stage 3 | 0.6327±**0.0087** | **0.0801**±**0.0359** |
| Adult | Stage 1 | **0.7778**±0.0530 | 0.0167±0.0212 |
| | Stage 2 | 0.7611±0.0450 | 0.0069±0.0094 |
| | Stage 3 | 0.7383±0.0206 | 0.0109±0.0102 |
| | Stage 1+Stage 2+Stage 3 | 0.7685±**0.0281** | **0.0036**±**0.0009** |
| eICU | Stage 1 | 0.6488±0.0223 | 0.0189±0.0263 |
| | Stage 2 | 0.6503±0.0207 | 0.0356±0.023 |
| | Stage 3 | 0.6446±0.0226 | 0.0309±0.0209 |
| | Stage 1+Stage 2+Stage 3 | **0.6530**±**0.0195** | **0.0209**±**0.0201** |

settings on EFFL, we use the Adult dataset as an example and set the values of $\epsilon_b$, $\epsilon_{vl}$ and $\epsilon_{vb}$ to $[0.01, 0.02, 0.05, 0.1]$ respectively.

The result in Tab. 5 shows that: (1) $\epsilon_b$ has the ability to control the decision bias of the global model, and the smaller the $\epsilon_b$, the lower the decision bias of the global model; (2) $\epsilon_{vl}$ and $\epsilon_{vb}$ have the ability to control the equality of the performance of the global model on the clients, specifically, the smaller the $\epsilon_{vl}$, the smaller the standard deviation of accuracy among the clients; the smaller the $\epsilon_{vb}$, the smaller the standard deviation of decision bias among the clients.

## D.5 ABLATION STUDY

The motivation for the three-stage algorithm is that, compared to solving the entire MCMOO problem in a single stage, the advantages of a multi-stage approach are as follows:❶ The staged solution facilitates a better balance of trade-offs: as we discussed in Sec.1, the main challenges in our work come from three types of trade-offs. Each stage is designed to solve the problem under one type of trade-off; ❷ By dividing the problem into three stages, we only need to focus on satisfying partial constraints in each stage, which is helpful for more effectively entering a more strictly constrained decision space in the following stage; ❸ The staged solution provides a higher convergence speed. In each stage, we only need to solve a subproblem, which can reduce the problem's complexity and improve the solution's speed.

In this experiment, we conducted an ablation study to evaluate the necessity of each stage within the three-stage algorithm, and the results are presented in Tab. 6. The results confirm that only the three-stage approach can obtain the desired hypothesis in the decision space defined by fairness constraints.

