# OpenReview forum: "EFFL: Egalitarian Fairness in Federated Learning for Mitigating Matthew Effect"
_ICLR.cc/2024/Conference — ICLR 2024 Conference Withdrawn Submission_

### Official Review · Reviewer_YyVq · 2023-10-31

**Soundness:** 2 fair
**Presentation:** 2 fair
**Contribution:** 2 fair
**Rating:** 3
**Confidence:** 3

**Summary:**

This paper investigates fairness in FL on both accuracy and decision bias. The authors propose a three-stage optimization method to optimize the proposed constrained multi-objective optimization problem. Numerical results show the benefits of the algorithm in terms of fairness.

**Strengths:**

1. Considering both accuracy and decision bias is promising and important to FL research.
2. The proposed multi-objective optimization problem is straightforward.

**Weaknesses:**

1. The proposed algorithm, as described in Algorithm 1 (line 6), necessitates unbiased gradients in each communication round. However, conventional federated learning algorithms typically employ local SGD, rendering the gradients hard to compute.
2. Some comments regarding related works appear to be unfair. For instance, in [1], a certain degree of performance inequality is permitted within an acceptable threshold, defined as the "fair area."
3. The motivation for dividing the optimization method into three stages is not well-explained. It remains unclear whether the algorithm's performance would change if the first two stages were eliminated.
4. The experiments conducted on real datasets involve only 2 or 11 clients. It is advisable to include cross-device scenarios with hundreds of clients for a more comprehensive analysis.

[1] Pan Z, Wang S, Li C, Wang H, Tang X, Zhao J. Fedmdfg: Federated learning with multi-gradient descent and fair guidance. In Proceedings of the AAAI Conference on Artificial Intelligence, volume 37, pp. 9364–9371, 2023.

**Questions:**

1. Could the authors elaborate on how to implement the proposed algorithm when employing local SGD, a common approach in most federated learning papers?

2. Could the authors provide more detailed explanations and perform ablation studies to justify the design choices for the three stages of the algorithm?

---

> ### Author Response · Authors · 2023-11-16
> **Response to Reviewer YyVq**
>
> We appreciate the reviewer's constructive comments.
>
> 1.  **To the Weakness 1**
>
>     We'd like to clarify that our primary focus lies in introducing the crucial concept of achieving egalitarian fairness within FL to mitigate the Matthew effect, emphasizing social ethics. Our proposed training mechanism can be operated within a parallel SGD framework, where clients provide gradients. Similar to existing fair FL work [1], the reason we conduct EFFL in this context is the gradient directions can help to balance the trade-off between the model performance and fairness requirements better. Achieving egalitarian fairness in local SGD-based FL, where clients only provide model parameters, could be an interesting direction for our future work.
>
> 2.  **To the Weakness 2**
>
>     We'd like to clarify that [1] proposed an FL mechanism to achieve the fairness that they defined as "the uniformity of performance across clients", which is consistent with our description in related work "To achieve equal accuracy across clients". However, in a non-ideal scenario, absolute fairness is a nontrivial, due to the trade-off between multiple objectives. Therefore, their fairness solution is within an acceptable region, rather than being strictly fair.
>
> 3.  **To the Weakness 3**
>
>     We would like to explain the motivation for three-stage algorithm:
>
>     - The staged solution facilitates a better balance of trade-offs: as we discussed in Sec.1, the main challenges in our work come from three types of trade-offs. Each stage is designed to solve the problem under one type of trade-off;
>
>     - By dividing the problem into three stages, we only need to focus on satisfying partial constraints in each stage, which is helpful for more effectively entering a more strictly constrained decision space in the following stage;
>
>     - The staged solution provides a higher convergence speed. In each stage, we only need to solve a subproblem, which can reduce the problem's complexity and improve the solution's speed.
>
>     We conduct an ablation study to examine the effect of each stage separately and present the results in the following. The results confirm that only the three-stage approach can obtain the desired hypothesis in the decision space defined by fairness constraints.
>
>      | Dataset | Stage 1 Acc. | Stage 1 Bias | Stage 2  Acc. | Stage 2 Bias | Stage 3  Acc. | Stage 3 Bias | Stage 1+Stage 2+Stage 3 Acc. | Stage 1+Stage 2+Stage 3 Bias |
>     |---------|-------------------|-------------------|-------------------|-------------------|-------------------|-------------------|----------------------------------|----------------------------------|
>     | Synthetic | 0.6519±0.0169 | 0.1033±0.0477 | 0.5740±0.0661 | 0.0972±0.0740 | **0.6840**±0.0983 | 0.2483±0.0670 | 0.6327±**0.0087** | **0.0801**±**0.0359** |
>     | Adult | **0.7778**±0.0530 | 0.0167±0.0212 | 0.7611±0.0450 | 0.0069±0.0094 | 0.7383±0.0306 | 0.0109±0.0102 | 0.7685±**0.0281** | **0.0036**±**0.0009** |
>     | eICU | 0.6488±0.0223 | 0.0289±0.0263 | 0.6503±0.0207 | 0.0356±0.0234 | 0.6446±0.0226 | 0.0309±0.0209 | **0.6530**±**0.0195** | **0.0209**±**0.0201** |
>
> 4.  **To the Weakness 4**
>
>     We'd like to elaborate on the scalability of EFFL under more clients as follows: Our EFFL training objective (Eq. (6)) explicitly and separately considers the decision bias and egalitarian fairness of each client. Therefore, our method can ensure fair performance when faced with a larger number of clients. As the number of clients increases, the decision space defined by decision bias and egalitarian fairness may become more constrained, which could result in a decrease in accuracy performance due to the trade-off between accuracy and fairness constraints. However, our method provides a Pareto optimal solution that balances the trade-offs.
>
> 5.  **To the Question 1**
>
>     Same to response 1.
>
> 6.  **To the Question 2**
>
>     Same to response 3.

---

> ### Author Response · Authors · 2023-11-20
> **Response to Reviewer YyVq**
>
> **To the Weakness 4**
> Following your suggestion, we have conducted additional experiments under the ACSPublicCoverage dataset [2], a SOTA  and complex tabular dataset currently available for fairness research. The dataset can be naturally partitioned into $51$ clients according to the State where the data was collected. The dataset was collected in the year 2022, and the results of the experiment are summarized as follows. Our method achieves superior performance in reducing decision bias and ensuring egalitarian fairness among clients. Ditto has the best accuracy but at the cost of high decision bias and significant performance disparity among clients, which may exacerbate the Matthew effect and be undesirable from a social welfare perspective.
>
>    | Method | Local Acc. Avg. | Local Acc. Std. | Local Bias Avg. | Local Bias Std. |
> |--------|-----------------|-----------------|-----------------|-----------------|
> | FedAvg | 0.5778          | 0.0352          | 0.0260          | 0.0242          |
> | q-FFL  | 0.5888          | 0.0406          | 0.0236          | 0.0202          |
> | Ditto  | **0.6578**      | 0.0584          | 0.0307 ↑        | 0.0328          |
> | FedMDFG| 0.5978          | 0.0453          | 0.0215          | 0.0187          |
> | FedAvg+FairBatch | 0.5972 | 0.0454          | 0.0214          | 0.0188          |
> | FedAvg+FairReg   | 0.5964 | 0.0453          | 0.0202          | 0.0194          |
> | FCFL   | 0.6085          | 0.0453          | 0.0215          | 0.0187          |
> | EFFL   | 0.6037          | **0.0284**      | **0.0147**      | **0.0128**      |
>
> [2] Ding, F., Hardt, M., Miller, J., & Schmidt, L. (2021). Retiring adult: New datasets for fair machine learning.

---

### Official Review · Reviewer_Raor · 2023-11-01

**Soundness:** 3 good
**Presentation:** 2 fair
**Contribution:** 3 good
**Rating:** 8
**Confidence:** 4

**Summary:**

This paper examines the problem of fairness in a federated learning setting through the lens of MCMOO. The goal is to reduce global loss while satisfying several constraints: (1) Roughly equal loss/accuracy for each individual client; (2) Fairness at each individual client, i.e., low local unfairness; and (3) Local unfairness at each client is also roughly equal to global which reduces to "equal" unfairness at each client, i.e., equal local unfairness. The MCMOO problem is addressed by breaking it down into a three-stage optimization process where some of the stages are constrained optimizations. The approach to solving MCMOO has several interesting ideas that involve cleverly controlling the direction of the gradient such that certain other metrics do not increase.
Experimental results are provided on a synthetic dataset, Adult (2 clients), and eICU (11 clients).

**Strengths:**

Some of these points have already been discussed in the Summary.
Formulating the problem as an MCMOO and then attempting to solve it as a three-stage process is a good contribution. The approach to solving MCMOO has several interesting concepts that involve cleverly controlling the direction of the gradient such that certain other metrics do not increase.
They have also provided experimental results with several baselines.

**Weaknesses:**

1. Such MCMOO problems are known to be quite difficult and unstable sometimes. Could the authors comment on the stability of their strategy?

2. One weakness is that there is no theoretical guarantee on why this algorithm will converge, and under what conditions would it diverge. Is it possible to formalize the intuitions of Fig. 2 into a theorem? Or, at least discuss some scenarios where it would diverge.
For instance, there are also impossibility results on group fairness in federated learning in: https://arxiv.org/abs/2307.11333

3. The word Egalitarian is a bit confusing here. I would think Egalitarian would mean equal accuracy/loss across clients. But, in addition to that, there is also a group fairness criterion with local and global fairness constraints. Using the word "group" fairness at places would make it more clear.

4. For what class of fairness metrics would the algorithm work? Please comment.

5. Could the authors also elaborate on the communication cost when there is a maximum since to obtain the maximum also one needs all the values? How is communication complexity improved?

Additional Limitations:
There is no discussion on the privacy of this approach in a federated context.

**Questions:**

Already included several questions along with the weaknesses

---

> ### Author Response · Authors · 2023-11-16
> **Response to Reviewer Raor**
>
> We appreciate  the reviewer for acknowledging the contributions of our work and valuable comments.
>
> 1.  **To the Weakness 1**
>
>     Given gradients from the clients have an error $\tilde G (\theta^t)=G (\theta^t)+\mathbf{e} ^t$, then $\left \| \tilde{\mathbf{\alpha} } -\mathbf{\alpha} \right \| _2\le \mathcal{O}(max_t\left \| e^t \right \|_2 )$  in the convex optimization of Eq. (14) . Following that, the error of the model parameter $\theta^{t+1}$ is bounded by:
> $\left \| \tilde{ \theta} ^{t+1}-\theta^{t+1} \right \| _2 =\eta\left \| \tilde{\alpha}  ^{T}{G}+\tilde{\alpha}  ^{T}e^t-{\alpha}  ^{T}{G}\right \| _2
>   \le \eta \left \| G \right \| _2\mathcal{O} \left(\max _{t}\left\|e^{t}\right\| _{2}\right)+\eta \mathcal{O}\left(\max _{t}\left\|e^{t}\right\| _{2}\right)\left \|e^t  \right \| _2+\left \|a  \right \|  _2\left \|e^t  \right \| _2$
>
>    Given that $\eta$, $\left \| G \right \| _2$, and $\left \|a  \right \|  _2$ are bounded, $ \left \| \tilde{ \theta} ^{t+1}-\theta^{t+1} \right \| _2\le \mathcal{O} \left(\max _{t}\left\|e^{t}\right\|\_{2}\right) $ is also bounded in our algorithm, the model's sensitivity to input errors satisfies $\mathcal{O} \left(\max _ {t}\left|e^{t}\right| _{2}\right)$-stability.
>
> 2.  **To the Weakness 2**
>
>        **Proposition.** For $N$ optimization objectives $l_1(\theta^t),...,l_N(\theta^t)  $ and the  model parameter updating rule: $\theta^{t+1}=\theta^{t}+\eta d$; if $d^T\nabla_{\theta}l_i\le0$, there exists $\eta_0$ such that for $\forall \eta \in [0,\eta_0]$, the objectives $ l_1(\theta^t),...,l_N(\theta^t) $ will not increase, and the iterations toward convergent.
>
>     **Proof.** Performing Taylor expansion at $\theta^t$, we have $l_i(\theta^{t+1})-l_i(\theta^{t})=\eta d^T\nabla_{\theta} l_i (\theta^t)+R_1(\theta^{t}+\eta d)$, where $R_1$ is a higher-order infinitesimal of $\eta $, denoted by $o(\eta )$ ($ R_1(\theta^{t}+\eta d)=\frac{\nabla ^2_{\theta}l(\theta^t)}{2!}\eta^2 d^2=o(\eta)$). Therefore, we have $l _i(\theta^{t+1})-l _i(\theta^{t})=\eta d^T\nabla _{\theta} l_i (\theta^t)+o(\eta)$, since $o(\eta)$ approaches $0$ faster than $\eta$, when $d^T\nabla _{\theta} l _i (\theta^t)\le 0$, there exists $\eta_0>0$ such that for $\forall \eta \in [0,\eta_0]$, $l_i(\theta^{t+1})-l_i(\theta^{t})\le 0$.
>
>     Based on the proposition, the parameters update in the proposed algorithm is towards a gradient descent direction $d$ that satisfies $d^T\nabla_{\theta}  g_k(h)\le 0$, where $g_k$ is the different optimization objective at different stages. Thus, the proposed algorithm is convergent.
>
> 3.  **To the Weakness 3**
>
>     We'd like to clarify the **egalitarian fairness** and **group fairness**. Group fairness focuses on the performance disparity among protected groups, while egalitarian fairness focuses on performance (**local accuracy** and **local decision bias**) disparity among clients. **Egalitarian fairness of the local decision bias** requires equal bias levels across all clients, different from local group fairness, targeting model bias within a **single** local client. Egalitarian fairness may resemble group fairness when a client exclusively presents one protected group. However, this is a rare case. Hence, we propose EFFL for broader scenarios.
>
> 4.  **To the Weakness 4**
>
>     The main requirement for the fairness metrics is differentiable, e.g., we compute the TPSD gradients on its differentiable surrogates,
>
>     $TPSD=\sqrt{\frac{ {\sum _{i=1}^{M}}\left ( \mathbb{E} _{(X=\mathbf{x} \mid S=i,Y=1)}\left[h _{\boldsymbol{\theta}}(\mathbf{x})\right]-\frac{\sum _{i=1}^{M}\mathbb{E} _{(X=\mathbf{x} \mid S=i,Y=1)}\left[h _{\boldsymbol{\theta}}(\mathbf{x})\right]}{M} \right ) ^2}{M} } $.
>
>     Other metrics, such as DP, EO, can be made differentiable in a similar way and applied to our algorithm. We choose TPSD/APSD because they are more general and can handle non-binary sensitive attributes. Moreover, our algorithm can be applied to non-binary target variables, e.g., by replacing the BCELoss with a multi-class loss function $loss=-\sum_{i=0}^{C-1} y_{i} \log \left(p_{i}\right)$, and replacing the decision bias metric with
>     $TPSD =max_{y\in[|Y|]}\sqrt{\frac{\sum_{i=1}^{M}\left (\operatorname{Pr}\left (\hat{Y}\_k=y \mid A_k=i, Y_k=y\right )-\mu\right )^{2}}{M}} $.
>
> 5.  **To the Weakness 5**
>
>     Similar to prior fair FL methods, our approach involves increased parameter communication compared to non-fair FL. Each client sends gradients of the local loss function and local decision bias, along with their evaluations, to the server. This roughly doubles communication compared to non-fair FL. However, server-to-client communication remains unchanged, limited to broadcasting model parameters. Despite the heightened communication, our method achieves Pareto optimal model performance under egalitarian fairness. The performance distribution variance, mean, maximum, etc., are computed and used internally within the server without extra communication costs.

---

> > ### Author Response · Authors · 2023-11-18
> > **Response to Reviewer Raor**
> >
> > 6. **To the additional limitation**
> >
> >     Both privacy and fairness hold significant societal considerations within FL. Our work's primary focus is not solving the trade-offs between privacy and fairness. Instead, our core contribution lies in identifying the type of fairness that FL needs to avoid the Matthew effect - egalitarian fairness -  and propose a novel method to address the unexplored issue. Moreover, our approach is compatible with some privacy-preserving mechanisms, such as differential privacy, which can limit the leakage of loss and gradient information. This opens up a promising future direction to combine both aspects and enhance the privacy of our method.   In our revision, we would like to add the privacy discussion as your suggestion.

---

### Official Review · Reviewer_hFqL · 2023-11-03

**Soundness:** 2 fair
**Presentation:** 2 fair
**Contribution:** 3 good
**Rating:** 6
**Confidence:** 2

**Summary:**

This paper studies fairness in federated learning with. Specifically, it considers two types of fairness. The first one is the Matthew effect, which considers the performance across different clients; and the second one is the decision bias, which is a class of commonly studied fairness definitions like accuracy parity or equal opportunity across different demographic groups. The authors formulate the problem as a multi-constrained multi-objective optimization problem and propose a 3-stage solution to gradually search the optimal hypothesis in a more constrained hypothesis space. Experiments on both synthetic data and real-world data demonstrate the effectiveness of the proposed method over baseline methods.

**Strengths:**

S1. Fair federated learning is a practical problem, and it is good to

S2. The 3-stage solution is interesting.

S3. Experimental results show the effectiveness on the tested datasets over baseline methods.

**Weaknesses:**

W1. The paper needs stronger motivation to support the need of two fairness considerations. Right now it feels more like two fairness considerations are both important, so we will consider them simultaneously. Is it possible to provide some real-world examples or use cases?

W2. Fig. 1 (a) is too hypothetical to support the claim that poor model could impair data generation capabilities and worsening the performance gap over time.

W3. Is there any trade-off or correlation between the Matthew effect and APSD/TPSD? For example, if we only improve the Matthew effect, the accuracy of poorer model might increase, and it may further help reduce APSD/TPSD. Is it possible to show empirical analysis about it (e.g., ablation study)?

W4. In Definition 1, $f_k(h)$ is constrained to be no larger than $\epsilon_b$, and $\{f_k(h) - {\bar f}(h)\}$ is also constrained to be less than $\epsilon_{vb}$. Is it possible that enforcing these two set of constraints could hurt a fair local client? Consider a case where the global model is almost perfectly fair, $f_k(h) \approx 0$, but violate $f_k(h) - {\bar f}(h) \leq \epsilon_{vb}$. Then it might be possible to increase $f_k(h)$ (i.e., making it more biased) to enforce $f_k(h) - {\bar f}(h) \leq \epsilon_{vb}$.

W5. In several places, the authors mention that existing works cannot narrow the gap between worse-performing clients and better-performing clients. I don't understand why the claim is true. For example, minimax-based solution will always minimize the worst-performing clients so naturally it could reduce the disparity among different clients' performance. Is it possible to elaborate the reason more clearly?

W6. It feels a bit conflicting that the authors say minimax cannot narrow the gap among clients' performance but still solve a minimax problem in stages 2 and 3.

W7. Is there any analysis to show the convergence of the proposed method? The authors simply claim it can obtain the convergent solution.

--- Post Rebuttal ---
I appreciate authors' efforts to address my cocerns, and I have updated my score.

**Questions:**

Please see weaknesses.

---

> ### Author Response · Authors · 2023-11-16
> **Response to Reviewer hFqL**
>
> We appreciate the reviewer's constructive comments. Our responses to the comments are listed as follows:
>
> 1.  **To the W1**
>
>     We'd like to explain that the motivation for two fairness is that clients may contribute varying amounts due to inherent resource inequalities. Ignoring such inequities can result in the accumulation of resource inequities due to the Matthew effect. This issue is particularly noticeable when FL is applied in social welfare scenarios, e.g., collaborative training of a disease diagnosis model among hospitals. Hospitals with lower data resources will obtain poorer models. Lower accuracy and trustworthy will affect their subsequent diagnosis, leading to continuous resource inequality and the deterioration of social welfare.
>
> 2.  **To the W2**
>
>     We thank the reviewer for the kind suggestion, and we will revise the figure better.
>
> 3.  **To the W3**
>
>     As we explained in Sec. 1, there is a trade-off between the Matthew effect and APSD/TPSD. Specifically, it manifests in the following way:
>
>     - Equal accuracy distribution is achieved by improving the model's fit towards disadvantaged clients rather than advantaged clients. As reducing the model's fit on a dataset can mitigate decision bias [1], APSD/TPSD may decrease on advantaged clients but increase on disadvantaged clients.
>     - Equal decision bias distribution and minimizing APSD/TPSD have a more intuitive trade-off; for clients with inherently larger APSD/TPSD, APSD/TPSD decrease, while for clients with inherently smaller APSD/TPSD, APSD/TPSD increase.
>
>     Therefore, the trade-off we need to address is how to reduce the Matthew effect by egalitarian fairness while ensuring that APSD/TPSD increasing on certain clients still remains within an acceptable threshold.
>
>     We conduct an ablation experiment on Adult dataset to verify the trade-off between egalitarian fairness and APSD/TPSD.
>
>     | Considered Fairness         | Client1 Acc. | Client1 TPSD | Client2 Acc. | Client2 TPSD |
>     | --------------------------- | ------------ | ------------ | ------------ | ------------ |
>     | None fairness               | 0.7348       | 0.0393       | 0.8186       | 0.0262       |
>     | Egalitarian fairness        | 0.7502       | 0.0242       | 0.8071       | 0.0399 ↑     |
>     | Egalitarian fairness + TPSD | 0.7403       | 0.0045       | 0.7967       | 0.0026       |
>
>     [1]Chouldechova, A., and Roth, A. (2020). A snapshot of the frontiers of fairness in machine learning.
>
> 4.  **To the W4**
>
>     We'd like to clarify that it's acceptable to increase a client's $f_k$ from $0$ to non-zero, as long as it does not violate the decision bias threshold $\epsilon_b$, in order to better achieve the egalitarian fairness objective and avoid the Matthew effect. Only pursuing the optimal performance of a single client $f_k\rightarrow0$, may result in poor performance of the model on other clients, which is undesirable from the perspective of social welfare and ethics.
>
> 5.  **To the W5**
>
>     The existing methods, e.g. min-max-based, only focus on improving the performance of the worst-performing client and may harm others; thus, we consider **egalitarian fairness**, which aims to achieve a more equal distribution of performance among all clients simultaneously.
>
> 6.  **To the W6**
>
>     We avoid the drawback of the original min-max by introducing non-positive gradient constraints on non-worst clients. This makes the modified min-max equivalent to multi-objective optimization, improving the worst client without harm to other clients.
>
> 7.  **To the W7**
>
>     We'd like to clarify the convergence of the proposed algorithm as follows:
>
>     **Proposition.** For $N$ optimization objectives $l_1(\theta^t),...,l_N(\theta^t)  $ and the  model parameter updating rule: $\theta^{t+1}=\theta^{t}+\eta d$; if $d^T\nabla_{\theta}l_i\le0$, there exists $\eta_0$ such that for $\forall \eta \in [0,\eta_0]$, the objectives $ l_1(\theta^t),...,l_N(\theta^t) $ will not increase, and the iterations toward convergent.
>
>     **Proof.** Performing Taylor expansion at $\theta^t$, we have $l_i(\theta^{t+1})-l_i(\theta^{t})=\eta d^T\nabla_{\theta} l_i (\theta^t)+R_1(\theta^{t}+\eta d)$, where $R _1$ is a higher-order infinitesimal of $\eta $, denoted by $o(\eta )$ ($ R_1(\theta^{t}+\eta d)=\frac{\nabla ^2 _{\theta}l(\theta^t)}{2!}\eta^2 d^2=o(\eta)$). Therefore, we have $l _i(\theta^{t+1})-l _i(\theta^{t})=\eta d^T\nabla  _{\theta} l_i (\theta^t)+o(\eta)$, since $o(\eta)$ approaches $0$ faster than $\eta$, when $d^T\nabla _{\theta} l_i (\theta^t)\le 0$, there exists $\eta_0>0$ such that for $\forall \eta \in [0,\eta_0]$, $l_i(\theta^{t+1})-l_i(\theta^{t})\le 0$.
>
>     Based on the proposition, the parameters update in the proposed algorithm is towards a gradient descent direction $d$ that satisfies $d^T\nabla_{\theta}  g_k(h)\le 0$, where $g_k$ is the different optimization objective at different stages. Thus, the proposed algorithm is convergent.

---

> > ### Comment · Reviewer_hFqL · 2023-11-23
> > **Thank you for the response**
> >
> > Thank you for the response. Most of my concerns are addressed, except for W2, because no revised version with updated figure is uploaded yet. For W2, it would be better to see an updated version of this manuscript. For other concerns, it would be good to incorporate such discussions in the revised version. I will keep my current score but am open to increase score if updated version is available.

---

> ### Author Response · Authors · 2023-11-23
> **Response to Reviewer hFqL**
>
> Thank you for your reply. We have submitted a revised version. Regarding W2, the figure was originally designed to illustrate how the Matthew effect influences the cycle of “data generation → model training → model deployment→ data generation”. However, to address W1, we have added an example of collaborative training of a diagnostic model  in the Sec. 1, which can help the readers better understand how the Matthew effect affects model training. Therefore, the original figure was no longer necessary, and we have removed it.

---

### Official Review · Reviewer_ta1u · 2023-11-09

**Soundness:** 2 fair
**Presentation:** 2 fair
**Contribution:** 2 fair
**Rating:** 5
**Confidence:** 4

**Summary:**

This work proposes EFFL a client fairness approach that aims to mitigate the Mathew effect by producing a Pareto optimal model with equal decision bias and accuracy across the participating clients. EFFL problem is formulated as a MOOP with decision bias and fairness constraints. The authors provide a 3-step algorithm for solving this objective and perform experiments that showcase superior performance compared to various other baselines on a synthetic dataset and 2 real datasets.

**Strengths:**

* Addressing the combinatorial issue of achieving equitable performance across clients in federated learning holds significant importance.

* The authors compared with multiple baselines and considered settings with adversarial attacks.

**Weaknesses:**

* The proposed algorithm is complicated and lacks formal convergence guarantees, so it's hard to understand and confidence in the algorithm's behaviour and optimality of the produced global model.
* It would be beneficial to have some guidance or a more systematic approach to determine the values of $\epsilon_b$, $\epsilon_{ub}$ and $\epsilon_{ul}$.
* The proposed problem and algorithms assume a binary target variable which is rather restrictive.
* The algorithm requires full client participation and allows for a single local epoch, increasing the communication overhead (as also shown in Figure 3) and restricting the applicability of EFFL in large-scale FL applications.
 * The experiments were conducted on only 2 real datasets and considered very few clients (maximum 11 for the eICU dataset).

**Questions:**

* How to determine the values of $\epsilon_b$ $\epsilon_{ub}$ and $\epsilon_{ul}$? There is some study on the effects of Appendix B.4.2, but I am unsure whether these results can be generalized given it was only examined for a single dataset that uses 2 clients.
 * Can the spaces required by the EFFL algorithm be infeasible? (e.g., the decision space $\mathcal{H}_B\cap\mathcal{H}_E$ defined by the fairness and bias constraints)
 * Are there any assumptions on the smoothness and convexity of the hypothesis class and the local loss functions to get the final objective?
* How well does this approach scale with a large number of clients? It would be interesting to see what EFFL's performance on the fe.g., on ACS Employment dataset, which naturally exhibits non-iid characteristics, being partitioned into 51 regions (that can act as separate clients).


**Minor:**
* what does the following sentence mean in the context of FL: "Previous work overlooks the trade-offs in achieving equality from a social welfare perspective and local optimality from an individual beneficial perspective"?
* [1] i missing from related work

[1]  Hu, S., Wu, Z. S., and Smith, V. (2022). Fair federated learning via bounded group loss.

---

> ### Author Response · Authors · 2023-11-16
> **Response to Reviewer ta1u**
>
> We appreciate the reviewer's constructive comments. Our responses to the comments are listed as follows:
>
> 1.  **To the Weakness 1**
>
>     We'd like to clarify the convergence of the proposed algorithm as follows:
>
>     **Proposition.** For $N$ optimization objectives $l_1(\theta^t),...,l_N(\theta^t)  $ and the  model parameter updating rule: $\theta^{t+1}=\theta^{t}+\eta d$; if $d^T\nabla_{\theta}l_i\le0$, there exists $\eta_0$ such that for $\forall \eta \in [0,\eta_0]$, the objectives $ l_1(\theta^t),...,l_N(\theta^t) $ will not increase, and the iterations toward convergent.
>
>     **Proof.** Performing Taylor expansion at $\theta^t$, we have $l_i(\theta^{t+1})-l_i(\theta^{t})=\eta d^T\nabla_{\theta} l_i (\theta^t)+R_1(\theta^{t}+\eta d)$, where $R_1$ is a higher-order infinitesimal of $\eta $, denoted by $o(\eta )$ ($ R_1(\theta^{t}+\eta d)=\frac{\nabla ^2_{\theta}l(\theta^t)}{2!}\eta^2 d^2=o(\eta)$). Therefore, we have $l_i(\theta^{t+1})-l_i(\theta^{t})=\eta d^T\nabla_{\theta} l_i (\theta^t)+o(\eta)$, since $o(\eta)$ approaches $0$ faster than $\eta$, when $d^T\nabla_{\theta} l_i (\theta^t)\le 0$, there exists $\eta_0>0$ such that for $\forall \eta \in [0,\eta_0]$, $l_i(\theta^{t+1})-l_i(\theta^{t})\le 0$.
>
>     Based on the proposition, the parameters update in the proposed algorithm is towards a gradient descent direction $d$ that satisfies $d^T\nabla_{\theta}  g_k(h)\le 0$, where $g_k$ is the different optimization objective at different stages. Thus, the proposed algorithm is convergent.
>
> 2.  **To the Weakness 2**
>
>     $\epsilon _b$,etc. are assigned based on the actual task requirements. We add the experiment under the eICU dataset (11 clients) as follows, to verify the scalability of the fairness budgets in controlling the fairness.
>     |  $\epsilon_b$ | 0.01 | 0.02 | 0.05 | 0.1 |
>     | --- | --- | --- | --- | --- |
>     | Avg. Acc. | 0.627 | 0.653 | 0.654 | **0.659** |
>     | Avg. Bias | **0.013** | 0.020 | 0.030 | 0.041 |
>
>     | $\epsilon_{vl}$ | 0.01      | 0.02  | 0.05  | 0.1   |
>     | --------------- | --------- | ----- | ----- | ----- |
>     | Std. Acc.       | **0.014** | 0.019 | 0.023 | 0.025 |
>
>     | $\epsilon_{vb}$ | 0.01      | 0.02  | 0.05  | 0.1   |
>     | --------------- | --------- | ----- | ----- | ----- |
>     | Std. Bias       | **0.011** | 0.020 | 0.032 | 0.036 |
>
> 3.  **To the Weakness 3**
>
>     The proposed algorithm can be applied to non-binary target variables, by replacing the BCELoss with a multi-class loss function, i.e., $loss=-\sum_{i=0}^{C-1} y_{i} \log \left(p_{i}\right)$, and replacing the decision bias metric with
>     $TPSD =max _{y\in[|Y|]}\sqrt{\frac{\sum _{i=1}^{M}\left (\operatorname{Pr}\left (\hat{Y} _k=y \mid A _k=i, Y _k=y\right )-\mu\right )^{2}}{M}}$.
>
> 4.  **To the Weakness 4**
>
>     Similar to prior works in fair FL, our algorithm involves additional parameters communication, which approximately doubles the communication cost compared to non-fair FL. The benefit of the additional communication cost is the Pareto optimal model performance under the egalitarian fairness constraints.
>
> 5.  **To the Weakness 5**
>
>     We'd like to clarify the scalability as the EFFL problem explicitly and separately considers the loss and fairness of each client. Therefore, our method is scalable when dealing with a larger number of clients. We appreciate your suggestion of ACS Employment dataset, and we'd like to incorporate it in future work.
>
> 6.  **To the Question 1**
>
>     Same to response 2.
>
> 7.  **To the Question 2**
>
>     We'd like to clarify that $\mathcal{H} _B\cap \mathcal{H} _E\neq \emptyset $, e.g., $\mathcal{H} _B\cap \mathcal{H} _E$ contains hypothesis $h\in \mathcal{H} _E$ with $\bar f(h)\le\epsilon _b-\epsilon _{vb}$. Since $h\in \mathcal{H} _E$, $h$ satisfies $| f _k \left ( h  \right ) - \bar{f} \left ( h \right )   | _{k=1}^N\le \epsilon _{vb}$, and thus, it can be readily deduced that $h$ also satisfies $f _k \left ( h \right ) _{k=1}^N \le \epsilon _b$ and is also contained within $\mathcal{H} _B$, which requires $f _k \left ( h \right ) _{k=1}^N \le \epsilon  _b$.
>
> 8.  **To the Question 3**
>
>     We'd like to clarify  EFFL can be applied to both convex and non-convex local loss functions.
>
> 9.  **To the Question 4**
>
>     Same to response 5.
>
> 10. **To the Minor 1**
>
>     Individual beneficial perspective means the model provides maximized individual performance but may harm  others. Social welfare perspective means limiting performance in advantaged clients to improve disadvantaged ones, which opposes individual beneficial perspective.
>
> 11. **To the Minor 2**
>
>     We'd like to add [1] to related work, which focuses on group-level fairness among protected groups. Our work focuses on client-level fairness, aiming at more uniform performance distribution among clients.

---

> ### Author Response · Authors · 2023-11-20
> **Response to Reviewer ta1u**
>
> **To the Weakness 5**
> Following your suggestion, we conducted experiments under the ACSPublicCoverage dataset, where $51$ clients are divided by the State where their data originated.  The dataset was collected in the year 2022, and the results of the experiment are summarized as follows. Our method achieves superior performance in reducing decision bias and ensuring egalitarian fairness among clients. Ditto has the best accuracy but at the cost of high decision bias and significant performance disparity among clients, which may exacerbate the Matthew effect and be undesirable from a social welfare perspective.
>
>    | Method | Local Acc. Avg. | Local Acc. Std. | Local Bias Avg. | Local Bias Std. |
> |--------|-----------------|-----------------|-----------------|-----------------|
> | FedAvg | 0.5778          | 0.0352          | 0.0260          | 0.0242          |
> | q-FFL  | 0.5888          | 0.0406          | 0.0236          | 0.0202          |
> | Ditto  | **0.6578**      | 0.0584          | 0.0307 ↑       | 0.0328          |
> | FedMDFG| 0.5978          | 0.0453          | 0.0215          | 0.0187          |
> | FedAvg+FairBatch | 0.5972 | 0.0454          | 0.0214          | 0.0188          |
> | FedAvg+FairReg   | 0.5964 | 0.0453          | 0.0202          | 0.0194          |
> | FCFL   | 0.6085          | 0.0453          | 0.0215          | 0.0187          |
> | EFFL   | 0.6037          | **0.0284**      | **0.0147**      | **0.0128**      |

---

> > ### Comment · Reviewer_ta1u · 2023-11-23
> >
> > I thank the authors for their responses. I have revised my score accordingly.

---

### Author Response · Authors · 2023-11-18
**Thank you!**

Thank you very much for the reviewers's efforts and valuable comments. We have provided the point-to-point response for each review in detail.  Should you have any further questions, do not hesitate to contact us.

---

> ### Author Response · Authors · 2023-11-23
> **Greatly appreciate your help!**
>
> We would like to thank all the reviewers for their thoughtful reviews and insightful suggestions. We are particularly glad that the reviewers recognized the significance of “equitable performance across clients” solved in our work: “holds significant importance”(Reviewer ta1u), and “considering both accuracy and decision bias is promising and important” (Reviewer YyVq). We also appreciate the reviewers’ positive feedback on our proposed algorithm: “3-stage solution is interesting” (Reviewer hFqL), “a three-stage process is a good contribution” (Reviewer Raor).
>
> We have made several improvements to our work and added some key experimental results that we believe substantially strengthen our paper in a revised version. In the revised version, the parts in red font are based on the common comments  of reviewers. The parts in green font are mainly based on Reviewer ta1u’s comments. The parts in blue font are based on Reviewer hFqL’s comments. The parts in purple font are based on Reviewer Raor’s comments. The parts in orange font are based on Reviewer YyVq’s comments.